# EFFICIENT WASSERSTEIN AND SINKHORN POLICY OPTIMIZATION

## ABSTRACT

Trust-region methods based on Kullback-Leibler divergence are pervasively used to stabilize policy optimization in reinforcement learning. In this paper, we examine two natural extensions of policy optimization with Wasserstein and Sinkhorn trust regions, namely *Wasserstein policy optimization (WPO)* and *Sinkhorn policy optimization (SPO)*. Instead of restricting the policy to a parametric distribution class, we directly optimize the policy distribution and derive their close-form policy updates based on the Lagrangian duality. Theoretically, we show that WPO guarantees a monotonic performance improvement, and SPO provably converges to WPO as the entropic regularizer diminishes. Experiments across tabular domains and robotic locomotion tasks further demonstrate the performance improvement of both approaches, more robustness of WPO to sample insufficiency, and faster convergence of SPO, over state-of-art policy gradient methods.

## 1 INTRODUCTION

Policy-based reinforcement learning (RL) approaches have received remarkable success in many domains, including video games (Mnih et al., 2013; Mnih et al., 2015), board games (Silver et al., 2016; Heinrich & Silver, 2016), robotics (Grudic et al., 2003; Gu et al., 2017), and continuous control tasks (Duan et al., 2016; Schulman et al., 2016). One prominent example is policy gradient method (Grudic et al., 2003; Peters & Schaal, 2006; Lillicrap et al., 2016; Sutton et al., 1999; Williams, 1992; Mnih et al., 2016; Silver et al., 2014). The core idea is to represent the policy with a probability distribution $\pi_\theta(a|s) = P[a|s; \theta]$, such that the action $a$ in state $s$ is chosen stochastically following the policy $\pi_\theta$ controlled by parameter $\theta$. Determining the right step size to update the policy is crucial for maintaining the stability of policy gradient methods: too conservative choice of stepsizes result in slow convergence, while too large stepsizes may lead to catastrophically bad updates.

To control the size of policy updates, Kullback-Leibler (KL) divergence is commonly adopted to measure the difference between two policies. For example, the seminal work on trust region policy optimization (TRPO) by Schulman et al. (2015) introduced KL divergence based constraints (trust region constraints) to restrict the size of the policy update; see also Peng et al. (2019); Abdolmaleki et al. (2018). Kakade (2001) and Schulman et al. (2017) introduced a KL-based penalty term to the objective to prevent excessive policy shift.

Though KL-based policy optimization has achieved promising results, it remains interesting whether using other metrics to gauge the similarity between policies could bring additional benefits. Recently, few work (Richemond & Maginnis, 2017; Zhang et al., 2018; Moskovitz et al., 2020; Pacchiano et al., 2020) has explored the Wasserstein metric to restrict the deviation between consecutive policies. Compared with KL divergence, the Wasserstein metric has several desirable properties. Firstly, it is a true symmetric distance measure. Secondly, it allows flexible user-defined costs between actions and is less sensitive to ill-posed likelihood ratios. Thirdly but most importantly, the Wasserstein metric has a weaker topology (Arjovsky et al., 2017), thus possibly leading to a more robust policy and more stable performance.

However, the challenge of applying the Wasserstein metric for policy optimization is also evident: evaluating the Wasserstein distance requires solving an optimal transport problem, which could be computationally expensive. To avoid this computation hurdle, existing work resorts to different techniques to *approximate the policy update* under Wasserstein regularization. For example, Richemond & Maginnis (2017) solved the resulting RL problem using Fokker-Planck equations;

Zhang et al. (2018) introduced particle approximation method to estimate the Wasserstein gradient flow. Recently, Moskovitz et al. (2020) instead considered the second-order Taylor expansion of Wasserstein distance based on Wasserstein information matrix to characterize the local behavioral structure of policies. Pacchiano et al. (2020) tackled behavior-guided policy optimization with smooth Wasserstein regularization by solving an approximate dual reformulation defined on reproducing kernel Hilbert spaces. Aside from such approximation, some of these work also limits the policy representation to a particular parametric distribution class, As indicated in Tessler et al. (2019), since parametric distributions are not convex in the distribution space, optimizing over such distributions results in local movements in the action space and thus leads to convergence to a sub-optimal solution. Henceforth, the theoretical performance of policy optimization under the Wasserstein metric remains elusive in light of these approximation errors.

In this paper, we study policy optimization with trust regions based on Wasserstein distance and Sinkhorn divergence. The latter is a smooth variant of Waserstein distance by imposing an entropic regularization to the optimal transport problem (Cuturi, 2013). We call them, *Wasserstein Policy Optimization (WPO)* and *Sinkhorn Policy Optimization (SPO)*, respectively. Instead of confining the distribution of policy to a particular distribution class, we work on the space of policy distribution directly, and consider all admissible policies that are within the trust regions with the goal of avoiding approximation errors. Unlike existing work, we focus on *exact characterization* of the policy updates. We highlight our contributions as follows:

1. **Algorithms:** We develop close-form expressions of the policy updates for both WPO and SPO based on the corresponding optimal Lagrangian multipliers of the trust region constraints. In particular, the optimal Lagrangian multiplier of SPO admits a simple form and can be computed efficiently. A practical on-policy actor-critic algorithm is proposed based on the derived expressions of policy updates and advantage value function estimation.

2. **Theory:** We theoretically show that WPO guarantees a *monotonic performance improvement* through the iterations, *even with non-optimal Lagrangian multipliers*, yielding better and more robust guarantee compared to that using KL divergence. Moreover, we prove that SPO converges to WPO as the entropic regularizer diminishes.

3. **Experiments:** A comprehensive evaluation with several types of testing environments including tabular domains and robotic locomotion tasks demonstrates the efficiency and effectiveness of WPO and SPO. Compared to state-of-art policy gradients approaches using KL divergence such as TRPO and PPO, our methods achieve better sample efficiency, faster convergence, and improved final performance. Our numerical study indicates that by properly choosing the weight of the entropic regularizer, SPO achieves a better trade-off between convergence and final performance than WPO.

*Related work:* Wasserstiein-like metrics have been explored in a number of works in the context reinforcement learning. Ferns et al. (2004) first introduced bisimulation metrics based on Wasserstein distance to quantify behavioral similarity between states for the purpose of state aggregation. Such bisimulation metrics were recently utilized for representation learning of RL; see e.g., Castro (2020); Agarwal et al. (2020). The most related work to ours are Richemond & Maginnis (2017); Zhang et al. (2018); Moskovitz et al. (2020); Pacchiano et al. (2020). These work directly use Wasserstein-like distance to measure proximity of policies instead of states. Unlike ours, these work apply Wasserstein distance as an explicit penalty function instead of trust-region constraints. Moreover, they use different strategies to approximate the Wasserstein distance. The only work that exploited Sinkhorn divergence in RL, to our best knowledge, is Pacchiano et al. (2020). In addition, few recent work has also exploited Wasserstein distance for imitation learning; see e.g., Xiao et al. (2019); Dadashi et al. (2021).

Wasserstsein-like metrics are also pervasively studied in distributionally robust optimization (DRO); see e.g., the survey by Kuhn et al. (2019) and references therein. Despite the similarity shared in the duality formulations, the DRO problems are fundamentally different from constrained policy optimization. We also point out that a recent concurrent work by Wang et al. (2021a) studied DRO using the Sinkhorn distance.

## 2 BACKGROUND AND NOTATIONS

**Markov Decision Process (MDP):** We consider an infinite-horizon discounted MDP, defined by the tuple $(\mathcal{S}, \mathcal{A}, P, r, \rho_0, \gamma)$, where $\mathcal{S}$ is the state space, $\mathcal{A}$ is the action space, $P : \mathcal{S} \times \mathcal{A} \times \mathcal{S} \rightarrow \mathbb{R}$ is the transition probability, $r : \mathcal{S} \times \mathcal{A} \rightarrow \mathbb{R}$ is the reward function, $\rho_0 : \mathcal{S} \rightarrow \mathbb{R}$ is the distribution of the initial state $s_0$, and $\gamma$ is the discount factor. We define the return of timestep $t$ as the accumulated discounted reward from $t$, $R_t = \sum_{k=0}^{\infty} \gamma^k r(s_{t+k}, a_{t+k})$, and the performance of a stochastic policy $\pi$ as $J(\pi) = \mathbb{E}_{s_0, a_0, s_1 \ldots}[\sum_{t=0}^{\infty} \gamma^t r(s_t, a_t)]$ where $a_t \sim \pi(a_t|s_t)$, $s_{t+1} \sim P(s_{t+1}|s_t, a_t)$. As shown in Kakade & Langford (2002), the expected return of a new policy $\pi'$ can be expressed in terms of the advantage over the old policy $\pi$: $J(\pi') = J(\pi) + \mathbb{E}_{s \sim \rho_v^{\pi'}, a \sim \pi'}[A^\pi(s, a)]$, where $A^\pi(s, a) = \mathbb{E}[R_t|s_t = s, a_t = a; \pi] - \mathbb{E}[R_t|s_t = s; \pi]$ represents the advantage function and $\rho_v^\pi$ represents the unnormalized discounted visitation frequencies with initial state distribution $v$, i.e., $\rho_v^\pi(s) = \mathbb{E}_{s_0 \sim v}[\sum_{t=0}^{\infty} \gamma^t P(s_t = s|s_0)]$.

**Trust Region Policy Optimization (TRPO):** In TRPO (Schulman et al., 2015), the policy $\pi$ is parameterized as $\pi_\theta$ with parameter vector $\theta$. For notation brevity, we use $\theta$ to represent the policy $\pi_\theta$. Then, the new policy $\theta'$ is found in each iteration to maximize the expected improvement $J(\pi') - J(\pi)$, or equivalently, the expected value of the advantage function:

$$\max_{\theta'} \quad \mathbb{E}_{s \sim \rho_v^\theta, a \sim \theta'}[A^\theta(s, a)]$$
$$\text{s.t.} \quad \mathbb{E}_{s \sim \rho_v^\theta}[d_{KL}(\theta', \theta)] \leq \delta, \tag{1}$$

where $d_{KL}$ represents the KL divergence and $\delta$ is the threshold of the distance between the new and the old policies. Note that here the expected value of the advantage function is an estimation as the visitation frequency $\rho_v^\theta$ is used rather than $\rho_v^{\theta'}$, which means the changes in state visitation frequencies caused by the changes in policy are ignored.

**Wasserstein Distance:** Given two probability distributions of policies $\pi$ and $\pi'$ on the discrete action space $\mathcal{A} = \{a_1, a_2, \ldots, a_N\}$, the Wasserstein distance between the policies is defined as:

$$d_W(\pi', \pi) = \inf_{Q \in \Pi(\pi', \pi)} \langle Q, M \rangle, \tag{2}$$

where $\langle \cdot, \cdot \rangle$ denotes the Frobenius inner product. The infimum is taken over all joint distributions $Q$ with marginals $\pi'$ and $\pi$, and $M$ is the cost matrix with elements $M_{ij} = d(a_i, a_j)$, where $d(a_i, a_j)$ is defined as the distance between actions $a_i$ and $a_j$.

**Sinkhorn Divergence:** Sinkhorn divergence (Cuturi, 2013) provides a smooth approximation of the Wasserstein distance by adding an entropic regularizer. The Sinkhorn divergence is defined as follows:

$$d_S(\pi', \pi|\lambda) = \inf_{Q \in \Pi(\pi', \pi)} \{\langle Q, M \rangle - \frac{1}{\lambda} h(Q)\}, \tag{3}$$

where $h(Q) = -\sum_{i=1}^{N} \sum_{j=1}^{N} Q_{ij} \log Q_{ij}$ represents the entropy term, and $\lambda > 0$ is a regularization parameter. Similarly, we use $Q^s$ to denote the joint distribution of $\pi(\cdot|s)$ and $\pi'(\cdot|s)$ with $\sum_{i=1}^{N} Q_{ij}^s = \pi(a_j|s)$ and $\sum_{j=1}^{N} Q_{ij}^s = \pi'(a_i|s)$. The intuition of adding the entropic regularization is: since most elements of the optimal joint distribution $Q$ will be 0 with a high probability, by trading the sparsity with entropy, a smoother and denser coupling between distributions can be achieved (Courty et al., 2014; 2016). Therefore, when the weight of the entropic regularization decreases (i.e., $\lambda$ increases), the sparsity of the divergence increases, and the Sinkhorn divergence converges to the Wasserstein metric, i.e., $\lim_{\lambda \to \infty} d_S(\pi', \pi|\lambda) = d_W(\pi', \pi)$. More critically, Sinkhorn divergence is useful to mitigate the computational burden of computing Wasserstein distance. In fact, the efficiency improvement that Sinkhorn divergence and the related algorithms brought paves the way to utilize Wassersterin-like metrics in many machine learning domains, including online learning (Cesa-Bianchi & Lugosi, 2006), model selection (Juditsky et al., 2008; Rigollet & Tsybakov, 2011), generative modeling (Genevay et al., 2018; Petzka et al., 2017; Patrini et al., 2019), dimensionality reduction (Huang et al., 2021; Lin et al., 2020; Wang et al., 2021b).

## 3  WASSERSTEIN POLICY OPTIMIZATION

Motivated by TRPO, here we consider a trust region based on the Wasserstein metric. Moreover, we lift the restrictive assumption that a policy has to follow a parametric distribution class by allowing all admissible policies. Then, the new policy $\pi'$ is found in each iteration to maximize the estimated expected value of the advantage function. Therefore, the *Wasserstein Policy Optimization* (WPO) framework is shown as follows:

$$
\begin{aligned}
\max_{\pi' \in \mathcal{D}} \quad & \mathbb{E}_{s \sim \rho_v^\pi, a \sim \pi'(\cdot|s)}[A^\pi(s,a)] \\
\text{where} \quad & \mathcal{D} = \{\pi' | \mathbb{E}_{s \sim \rho_v^\pi}[d_W(\pi'(\cdot|s), \pi(\cdot|s))] \le \delta\},
\end{aligned}
\tag{4}
$$

where the Wasserstein distance $d_W(\cdot, \cdot)$ is defined in (2).

In most practical cases, the reward $r$ is bounded and correspondingly, the accumulated discounted reward $R_t$ is bounded. So without loss of generality, we can make the following assumption:

**Assumption 1.** *Assume $A^\pi(s,a)$ is bounded, i.e., $\sup_{a \in \mathcal{A}, s \in \mathcal{S}} |A^\pi(s,a)| \le A^{max}$ for some $A^{max} > 0$.*

With Wasserstein metric based trust region constraint, we are able to derive the closed-form of the policy update shown in Theorem 1. The main idea is to form the Lagrangian of the constrained optimization problem presented above, and the detailed proof can be found in Appendix A.

**Theorem 1.** *(Closed-form policy update) Let $k_s^\pi(\beta, j) = argmax_{k=1...N}\{A^\pi(s, a_k) - \beta M_{kj}\}$, where $M$ denotes the cost matrix. If Assumption 1 holds, then an optimal solution to the WPO problem in (4) is given by:*

$$
\pi^*(a_i|s) = \sum_{j=1}^{N} \pi(a_j|s) f_s^*(i,j),
\tag{5}
$$

*where $f_s^*(i,j) = 1$ if $i = k_s^\pi(\beta^*, j)$ and $f_s^*(i,j) = 0$ otherwise, and $\beta^*$ is an optimal Lagrangian multipler corresponds to the following dual formulation:*

$$
\min_{\beta \ge 0} F(\beta) = \min_{\beta \ge 0} \left\{ \beta\delta + \mathbb{E}_{s \sim \rho_v^\pi} \sum_{j=1}^{N} \pi(a_j|s)[A^\pi(s, a_{k_s^\pi(\beta,j)}) - \beta M_{k_s^\pi(\beta,j)j}] \right\}.
\tag{6}
$$

*Moreover, we have $\beta^* \le \bar{\beta} := \max_{s \in \mathcal{S}, k, j=1...N, k \ne j} (M_{kj})^{-1}(A^\pi(s, a_k) - A^\pi(s, a_j))$.*

The exact policy update for WPO in (5) requires computing the optimal Lagrangian multiplier $\beta^*$ by solving the one-dimensional subproblem (6). A closed form of $\beta^*$ is not easy to obtain in general, except for special cases of the distance $d(x,y)$ or cost matrix $M$. In Appendix G, we provide the closed form of $\beta^*$ for the case when $d(x,y) = 0$ if $x = y$ and 1 otherwise.

**WPO Policy Update:**  Based on Theorem 1, we introduce the following WPO policy updating rule:

$$
\pi_{t+1}(a_i|s) = \mathbb{F}^{\text{WPO}}(\pi_t) := \sum_{j=1}^{N} \pi_t(a_j|s) f_s^t(i,j),
\tag{WPO}
$$

where we choose an arbitrary $k_s^{\pi_t}(\beta_t, j) \in argmax_{k=1,...,N}\{A^{\pi_t}(s, a_k) - \beta_t M_{kj}\}$ and set $f_s^t(k_s^{\pi_t}(\beta_t, j), j) = 1$ and other entries to be 0.

Note that different from (5), we allow $\beta_t$ to be chosen arbitrarily and time dependently. We show that such policy update always leads to a monotonic improvement of the performance even when $\beta_t$ is not the optimal Lagrangian multiplier. In particular, we propose two efficient strategies to update the multiplier $\beta_t$:

(i) Time-dependent $\beta_t$: To improve the computational efficiency, we can simply treat $\beta_t$ as a time-dependent control parameter, e.g., we can choose $\beta_t$ to be a diminishing sequence.

(ii) Approximation of optimal $\beta_t$: To improve the convergence, we can approximately solve the optimal Lagrangian multiplier based on Sinkhorn divergence. We will discuss this in more detail in Section 4.

Next, we provide theoretical justification that WPO policy update is always guaranteed to improve the true performance $J$ monotonically if we have access to the true advantage function. If the advantage function can only be evaluated inexactly with limited samples, then an extra estimation error will incur. The detailed proof can be found in Appendix B.

**Theorem 2.** *(**Performance improvement**) For any initial state distribution $\mu$ and any $\beta_t \geq 0$, if $||\hat{A}^\pi - A^\pi||_\infty \leq \epsilon$ for some $\epsilon > 0$, the WPO policy update with the inaccurate advantage function $\hat{A}^\pi$, guarantees the following performance improvement bound,*

$$J(\pi_{t+1}) \geq J(\pi_t) + \beta_t \mathbb{E}_{s \sim \rho_\mu^{\pi_{t+1}}} \sum_{j=1}^N \pi_t(a_j|s) M_{\hat{k}_s^{\pi_t}(\beta_t,j)j} - \frac{2\epsilon}{1-\gamma}. \tag{7}$$

Note that when the estimation error $\epsilon = 0$, we have a monotonic performance improvement $J(\pi_{t+1}) \geq J(\pi_t)$ for any $\beta_t \geq 0$. If the second term in (7) is non-zero, then we have a strict monotonic improvement. Compared to the performance bound when using KL-based trust region $J(\pi_{t+1}) \geq J(\pi_t) - \frac{2\epsilon}{1-\gamma}$ (see, e.g., Schulman et al. (2015); Cen et al. (2020)), using the Wasserstein metric yields a tighter performance improvement bound and is more robust to the choice of $\beta_t$.

## 4 SINKHORN POLICY OPTIMIZATION

In this section, we introduce Sinkhorn policy optimization (SPO) by constructing trust region with Sinkhorn divergence. In the following theorem, we derive the optimal policy update in each step when using Sinkhorn divergence based trust region. The proof follows a similar procedure as the Wasserstein policy optimization framework by forming the Lagrangian of the constrained optimization problem. Details are provided in Appendix C.

**Theorem 3.** *If Assumption 1 holds, then the optimal solution to the trust region constrained problem (4) with Sinkhorn divergence is:*

$$\pi_\lambda^*(a_i|s) = \sum_{j=1}^N \frac{\exp\left(\frac{\lambda}{\beta_\lambda^*} A^\pi(s, a_i) - \lambda M_{ij}\right)}{\sum_{k=1}^N \exp\left(\frac{\lambda}{\beta_\lambda^*} A^\pi(s, a_k) - \lambda M_{kj}\right)} \pi(a_j|s), \tag{8}$$

*where $M$ denotes the cost matrix and $\beta_\lambda^*$ is an optimal solution to the following dual formulation:*

$$\min_{\beta \geq 0} F_\lambda(\beta) = \min_{\beta \geq 0} \Big\{ \beta\delta - \mathbb{E}_{s \sim \rho_\upsilon^\pi} \sum_{j=1}^N \pi(a_j|s)(\frac{\beta}{\lambda} + \frac{\beta}{\lambda}\ln(\pi(a_j|s)) -$$

$$\frac{\beta}{\lambda}\ln[\sum_{i=1}^N \exp\left(\frac{\lambda}{\beta} A^\pi(s, a_i) - \lambda M_{ij}\right)]) + \mathbb{E}_{s \sim \rho_\upsilon^\pi} \sum_{i=1}^N \sum_{j=1}^N \frac{\beta}{\lambda} \frac{\exp\left(\frac{\lambda}{\beta} A^\pi(s, a_i) - \lambda M_{ij}\right) \cdot \pi(a_j|s)}{\sum_{k=1}^N \exp\left(\frac{\lambda}{\beta} A^\pi(s, a_k) - \lambda M_{kj}\right)} \Big\}. \tag{9}$$

*Moreover, we have $\beta_\lambda^* \leq \frac{2A^{max}}{\delta}$.*

In contrast to the Wasserstein dual formulation (6), the objective in the Sinkhorn dual formulation (9) is differentiable in $\beta$ and admits closed-form gradients (shown in Appendix E). With this gradient information, we can use gradient-based global optimization algorithms (Wales & Doye, 1998; Zhan et al., 2006; Leary, 2000) to find a global optimal solution $\beta_\lambda^*$ to (9).

Next, we show that if the entropic regularization parameter $\lambda$ is large enough, then the optimal solution $\beta_\lambda^*$ is a close approximation to the optimal solution $\beta^*$ to the Wasserstein dual formulation. The detailed proof is provided in Appendix F.

**Theorem 4.** *Define $\beta_{UB} = \max\{\frac{2A^{max}}{\delta}, \bar{\beta}\}$. The following holds:*

1. *$F_\lambda(\beta)$ converges to $F(\beta)$ uniformly on $[0, \beta_{UB}]$,*

2. *$\lim_{\lambda \to \infty} \arg\min_{0 \leq \beta \leq \beta_{UB}} F_\lambda(\beta) \subseteq \arg\min_{0 \leq \beta \leq \beta_{UB}} F(\beta)$.*

Although it is difficult to obtain the exact value of the optimal solution $\beta^*$ to the Wasserstein dual formulation (6), the above theorem suggests that we can approximate $\beta^*$ via $\beta_\lambda^*$ by setting up a relative large $\lambda$. In practice, we can also adopt a smooth homotopy approach by setting an increasing sequence $\lambda_t$ for each iteration and letting $\lambda_t \to \infty$.

**SPO Policy Update:** Based on Theorem 3, we introduce the following SPO policy updating rule:

$$\pi_{t+1}(a_i|s) = \mathbb{F}^{\text{SPO}}(\pi_t) := \sum_{j=1}^{N} \frac{\exp\left(\frac{\lambda_t}{\beta_t} A^{\pi_t}(s, a_i) - \lambda_t M_{ij}\right)}{\sum_{k=1}^{N} \exp\left(\frac{\lambda_t}{\beta_t} A^{\pi_t}(s, a_k) - \lambda_t M_{kj}\right)} \pi_t(a_j|s), \qquad \text{(SPO)}$$

Here $\lambda_t \geq 0$ and $\beta_t \geq 0$ are some control parameters. The parameter $\beta_t$ can be either computed via solving the one-dimensional subproblem (9) or simply set as a diminishing sequence.

## 5 A PRACTICAL ALGORITHM

In practice, the advantage value functions are often estimated from sampled trajectories. In this section, we provide a practical on-policy actor-critic algorithm, described in Algorithm 1, that combines WPO/SPO with advantage function estimation.

In each iteration of Algorithm 1, the first step is to collect trajectories, which can be either complete or partial. The difference is whether the return is considered thoroughly to the end of a planning horizon. If the trajectory is complete, the total return can be directly expressed as the accumulated discounted rewards $R_t = \sum_{k=0}^{T-t-1} \gamma^k r_{t+k}$. If the trajectory is partial, it can be estimated by applying the multi-step temporal difference (TD) methods (De Asis et al., 2017): $\hat{R}_{t:t+n} = \sum_{k=0}^{n-1} \gamma^k r_{t+k} + \gamma^n V(s_{t+n})$. Then for the advantage estimation, we can use Monte Carlo advantage estimation, i.e., $\hat{A}_t^{\pi_k} = R_t - V_{\psi_k}(s_t)$ or Generalized Advantage Estimation (GAE) (Schulman et al., 2016), which provides a more explicit control over the bias-variance trade-off. In the value update step, we use a neural net to rep-

---

**Algorithm 1:** On-policy WPO/SPO algorithm

---

Input: number of iterations $K$, learning rate $\alpha$
Initialize policy $\pi_0$ and value network $V_{\psi_0}$ with
  random parameter $\psi_0$
**for** $k = 0, 1, 2 \ldots K$ **do**
  Collect a set of trajectories $\mathcal{D}_k$ on policy $\pi_k$
  For each timestep $t$ in each trajectory,
    compute total returns $G_t$ and estimate
    advantages $\hat{A}_t^{\pi_k}$
  Update value:
  $\psi_{k+1} \leftarrow \psi_k - \alpha \nabla_{\psi_k} \sum (G_t - V_{\psi_k}(s_t))^2$
  Update policy:
  $\pi_{k+1} \leftarrow \mathbb{F}(\pi_k)$ via WPO or SPO with $\hat{A}_t^{\pi_k}$
**end**

---

resent the value function, where $\psi$ is the parameter that specifies the value net $s \to V(s)$. Then, we can update $\psi$ by using gradient descent, which significantly reduces the computational burden of computing advantage directly.

## 6 EXPERIMENTS

In this section, we evaluate the proposed WPO and SPO approaches on tabular domains and robotic locomotion tasks as presented in Algorithm 1. We compare the performance of our proposed methods with several benchmarks, including TRPO (Schulman et al., 2015), PPO (Schulman et al., 2017), and A2C (Mnih et al., 2016)[1]. We compare with A2C because it is similar to our framework in the sense that both of them are simple on-policy actor-critic methods that utilize the advantage information to perform policy updates. For environments with a discrete state space (e.g., tabular domains), policy updates are performed for all states at each iteration. For environments with a continuous state space, a random subset of states is sampled at each iteration to perform policy updates.

### 6.1 ABLATION STUDY

In this experiment, we first examine the sensitivity of WPO in terms of different strategies of $\beta_t$. We test four settings of $\beta$ value for WPO policy update: (1) Setting 1: computing optimal $\beta$ value for all policy update; (2) Setting 2: computing optimal $\beta$ value for first 20% of policy updates and decaying $\beta$ for the remaining; (3) Setting 3: computing optimal $\beta$ value for first 20% of policy updates and fix $\beta$ as its last updated value for the remaining; (4) Setting 4: decaying $\beta$ for all policy updates (e.g., $\beta_t = \frac{1}{t^2}$). In particular, Setting 3 is rooted in the observation that $\beta^*$ does not change significantly

---

[1]We use the implementations of TRPO, PPO and A2C from OpenAI Baselines (Dhariwal et al., 2017) for MuJuCo tasks and Stable Baselines (Hill et al., 2018) for other tasks.

throughout all the policy updates, especially in the later stage in the experiments carried out in the paper. Small perturbations are added to the approximate values to avoid any stagnation in updating. Taxi task (Dietterich, 1998) from tabular domain is selected for this experiment.

The performance comparisons and average run times are shown in Figure 1 and Table 1 respectively. Figure 1 and Table 1 clearly indicate a tradeoff between computation efficiency and accuracy in terms of different choices of $\beta$ value. Setting 2 is the most effective way to balance the tradeoff between performance and run time. For the rest of experiments, we adopt this setting for both WPO and SPO. We also compare WPO with SPO under different constant and time varying $\lambda$ values on the Taxi task. As shown in Figure 1, SPO converges faster than WPO. With more weight on the entropic regularization of Sinkhorn divergence (i.e., smaller $\lambda$), SPO can

Table 1: Run time comparison for different $\beta$ settings

| Runtime | Taxi (s) | CartPole (s) |
|---|---|---|
| Setting 1 | 1224 | 130 |
| Setting 2 | 648 | 63 |
| Setting 3 | 630 | 67 |
| Setting 4 | 522 | 44 |

speed up its convergence more; while as $\lambda$ increases, the convergence becomes slower but the final performance of SPO improves and becomes closer to the final performance of WPO, which verifies the convergence property of Sinkhorn to Wasserstein distance shown in Theorem 4. Therefore, the choice of $\lambda$ can effectively adjust the trade-off between convergence and final performance. With a proper $\lambda$ choice, SPO is able to attain a faster convergence speed with an optimum that is only slightly lower than WPO.

More experiments for ablation study is conducted on the Chain (Dearden et al., 1998) and CartPole (Barto et al., 1983) tasks. Results are provided in Appendix H.

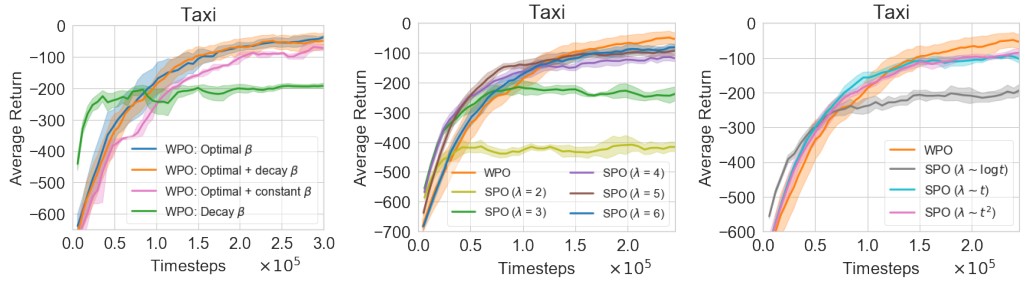

Figure 1: Episode rewards during the training process for the Taxi task with different $\beta$ and $\lambda$ settings, averaged across 3 runs with a random initialization. The shaded area depicts the mean $\pm$ the standard deviation.

## 6.2 TABULAR DOMAINS

We evaluate WPO and SPO on a set of tasks including Taxi, Chain (Dearden et al., 1998), and Cliff Walking (Sutton & Barto, 2018), which are intentionally designed to test the exploration ability of the algorithms. The tabular domain has a special environment structure with a discrete state space and a discrete action space. Thus, we use an array of size $|\mathcal{S}| \times |\mathcal{A}|$ to represent the policy $\pi(a|s)$. For the value function, we use a neural net to smoothly update the values. The performance of WPO and SPO are compared to the performance of TRPO, PPO and A2C under the same neural net structure. Each algorithm is evaluated 5 times with a random initialization. Results are reported in Table 2 and Figure 2. The setting of hyperparamaters and network sizes of our algorithms and additional results are provided in Appendix H.

Table 2: Trained agents performance on Taxi (averaged over 1000 episodes)

| | WPO | TRPO |
|---|---|---|
| Success (+20) | 0.753 | 0 |
| Fail (-10) | 0.232 | 0 |
| Timesteps (-1) | 70.891 | 200 |
| Avg Return | -58.151 | -200 |

As shown in Figure 2, the performances of WPO, SPO and TRPO are manifestly better than A2C and PPO. Between the trust region based methods, WPO and SPO outperform TRPO in most tasks, except in Chain, where the performances of these three methods are not significantly different. In Taxi and Cliff Walking, SPO converges to the optimum the fastest, while in Taxi, WPO converges to the best optimum, among all methods. We further analyze the performance of the trained agent for each algorithm on the Taxi environment. As shown in Table 2, WPO has a higher successful drop-off

rate and a lower task completion time while the original TRPO reaches the time limit with a drop-off rate 0. Therefore, the results in Taxi show that WPO finds a better policy than the original TRPO.

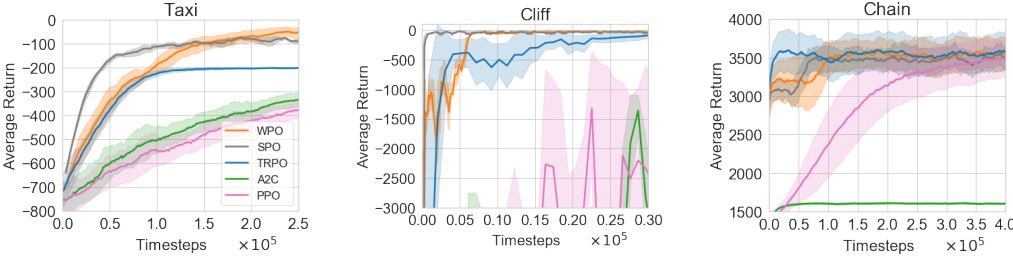

Figure 2: Episode rewards during the training process for the tabular domain tasks, averaged across 5 runs with a random initialization. The shaded area depicts the mean $\pm$ the standard deviation.

We also show that compared with the KL divergence, which is used in traditional TRPO and PPO approaches, the utilization of Wasserstein metric can cope with the inaccurate advantage function estimations caused by the lack of samples. We compare WPO with KL (Algorithm 1 framework with KL based policy update derived in Peng et al. (2019)) on the Chain task. We evaluate the performance of these two algorithms under different $N_A$, which denotes the number of samples used to estimate the advantage function at each iteration. As shown in Figure 3, when $N_A$ is 1000, KL performs slightly better than WPO. However, when $N_A$ decreases to 100 or 250, WPO outperforms KL. These results indicate that WPO is more robust than KL under inaccurate advantage values.

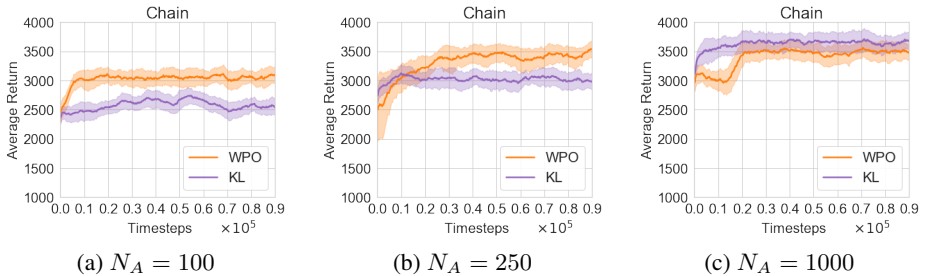

(a) $N_A = 100$        (b) $N_A = 250$        (c) $N_A = 1000$

Figure 3: Episode rewards during the training process for the Chain task, averaged across 3 runs with a random initialization. The shaded area depicts the mean $\pm$ the standard deviation.

## 6.3 ROBOTIC LOCOMOTION TASKS

We then integrate deep neural network architecture into MPO and SPO and evaluate their performance on several discrete locomotion tasks, including CartPole (Barto et al., 1983) and Acrobot (Geramifard et al., 2015). We use two separate neural nets to represent the policy and the value. The policy neural net receives state $s$ as an input and outputs the categorical distribution of $\pi(a|s)$. The performance of WPO and SPO are compared to that of TRPO, PPO and A2C under the same neural net structure. We run each algorithm 5 times with a random initialization.

**Final Performance:** Figure 4 shows the episode rewards during training process for WPO, SPO and baseline algorithms. As seen in Figure 4, WPO and SPO outperform TRPO, PPO and A2C in most tasks in terms of final performance, except in Acrobot where PPO performs the best. In most cases, SPO converges faster but WPO has a better final performance.

**Training Time:** To train $10^5$ timesteps in the discrete locomotion tasks, the training wall-clock time is around 63s for WPO, 65s for SPO, 59s for TRPO and 70s for PPO. Therefore, WPO has a similar computational efficiency as TRPO and PPO.

The performances of WPO and KL are also compared for the discrete locomotion tasks under different $N_A$. As shown in Figure 5, when $N_A$ is 500, KL performs better than WPO. However, when $N_A$ decreases to 100, WPO significantly outperforms KL. These results indicate that for discrete locomotion tasks, WPO is more robust than KL when advantage values are inaccurate.

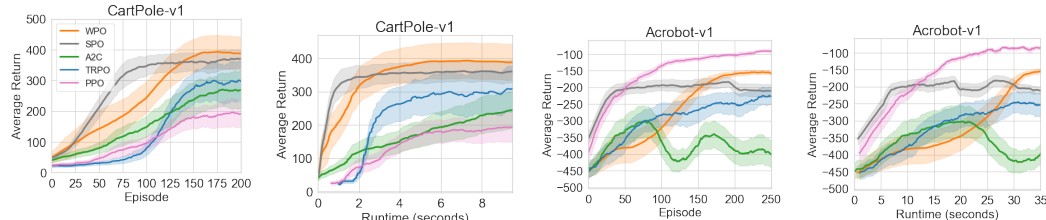

Figure 4: Episode rewards during the training process for the locomotion tasks, averaged across 5 runs with a random initialization. The shaded area depicts the mean ± the standard deviation.

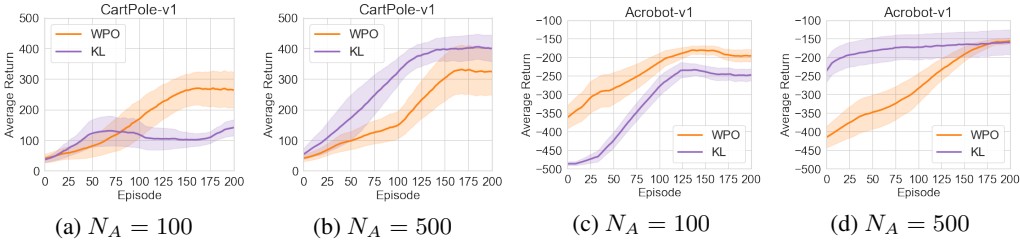

(a) $N_A = 100$      (b) $N_A = 500$      (c) $N_A = 100$      (d) $N_A = 500$

Figure 5: Episode rewards during the training process for the locomotion tasks, averaged across 3 runs with a random initialization. The shaded area depicts the mean ± the standard deviation.

## 6.4 CONTINUOUS ACTION SPACE:

We further extend the evaluation of WPO and SPO to environments with a continuous action space by discretizing the action space following Tang & Agrawal (2020). For comparison, we additionally consider Behavior Guided Policy Gradient (BGPG) algorithm from Pacchiano et al. (2020). Similar results are observed in Figure 6 as the discrete action tasks: WPO and SPO outperform the benchmark algorithms in terms of final performance.

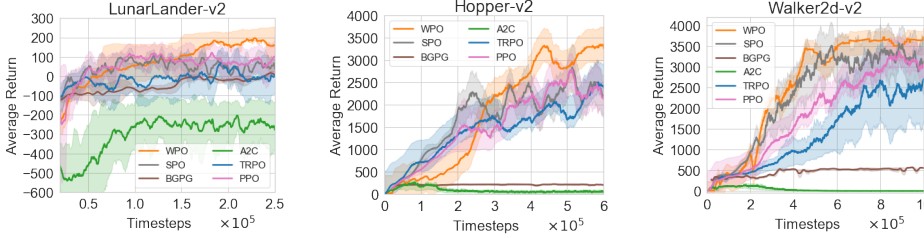

Figure 6: Episode rewards during the training process for continuous action space tasks, averaged across 3 runs with a random initialization. The shaded area depicts the mean ± the standard deviation.

## 7 CONCLUSION

In this paper, we present two policy optimization frameworks, WPO and SPO, which can exactly characterize the policy updates instead of confining their distributions to particular distribution class or requiring any approximation. Our methods outperform TRPO and PPO with better sample efficiency, faster convergence, and improved final performance. Our numerical results show that the Wasserstein metric is more robust to the ambiguity of advantage functions, compared with the KL divergence. Our strategy for adjusting $\beta$ value for WPO can reduce the computational time and boost the convergence without noticeable performance degradation. SPO improves the convergence speed of WPO by properly choosing the weight of the entropic regularizer. For future work, it remains interesting to extend the idea to PPO and natural policy gradients, which penalize the policy update instead of imposing trust region constraint, and extend it to off-policy frameworks.

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

# A  PROOF OF THEOREM 1

**Theorem 1.** *(Closed-form policy update)* Let $k_s^{\pi}(\beta, j) = argmax_{k=1\dots N}\{A^{\pi}(s, a_k) - \beta M_{kj}\}$, where $M$ denotes the cost matrix. If Assumption 1 holds, then an optimal solution to the WPO problem in (4) is given by:

$$\pi^*(a_i|s) = \sum_{j=1}^{N} \pi(a_j|s) f_s^*(i, j), \tag{5}$$

where $f_s^*(i, j) = 1$ if $i = k_s^{\pi}(\beta^*, j)$ and $f_s^*(i, j) = 0$ otherwise, and $\beta^*$ is an optimal Lagrangian multipler corresponds to the following dual formulation:

$$\min_{\beta \geq 0} F(\beta) = \min_{\beta \geq 0} \left\{ \beta\delta + \mathbb{E}_{s \sim \rho_v^{\pi}} \sum_{j=1}^{N} \pi(a_j|s)[A^{\pi}(s, a_{k_s^{\pi}(\beta, j)}) - \beta M_{k_s^{\pi}(\beta, j)j}] \right\}. \tag{6}$$

*Moreover, we have* $\beta^* \leq \bar{\beta} := \max_{s \in \mathcal{S}, k, j=1\dots N, k \neq j} (M_{kj})^{-1}(A^{\pi}(s, a_k) - A^{\pi}(s, a_j)).$

*Proof of Theorem 1.* First, we denote $Q^s$ as the joint distribution of $\pi(\cdot|s)$ and $\pi'(\cdot|s)$ with $\sum_{i=1}^{N} Q_{ij}^s = \pi(a_j|s)$ and $\sum_{j=1}^{N} Q_{ij}^s = \pi'(a_i|s)$. Also, let $f_s(i, j)$ represent the conditional distribution of $\pi'(a_i|s)$ under $\pi(a_j|s)$. Then $Q_{ij}^s = \pi(a_j|s)f_s(i, j)$, $\pi'(a_i|s) = \sum_{j=1}^{N} Q_{ij}^s = \sum_{j=1}^{N} \pi(a_j|s)f_s(i, j)$. In addition:

$$d_W(\pi'(\cdot|s), \pi(\cdot|s)) = \min_{Q_{ij}^s} \sum_{i=1}^{N}\sum_{j=1}^{N} M_{ij}Q_{ij}^s = \min_{f_s(i,j)} \sum_{i=1}^{N}\sum_{j=1}^{N} M_{ij}\pi(a_j|s)f_s(i, j), \text{ and}$$

$$\mathbb{E}_{a \sim \pi'(\cdot|s)}[A^{\pi}(s, a)] = \sum_{i=1}^{N} A^{\pi}(s, a_i)\pi'(a_i|s) = \sum_{i=1}^{N}\sum_{j=1}^{N} A^{\pi}(s, a_i)\pi(a_j|s)f_s(i, j).$$

Thus, the WPO problem in (4) can be reformulated as:

$$\max_{f_s(i,j) \geq 0} \quad \mathbb{E}_{s \sim \rho_v^{\pi}} \sum_{i=1}^{N}\sum_{j=1}^{N} A^{\pi}(s, a_i)\pi(a_j|s)f_s(i, j) \tag{10a}$$

$$s.t. \quad \mathbb{E}_{s \sim \rho_v^{\pi}} \sum_{i=1}^{N}\sum_{j=1}^{N} M_{ij}\pi(a_j|s)f_s(i, j) \leq \delta, \tag{10b}$$

$$\sum_{i=1}^{N} f_s(i, j) = 1, \qquad \forall s \in \mathcal{S}, j = 1\dots N. \tag{10c}$$

Note here that (10b) is equivalent to $\mathbb{E}_{s \sim \rho_v^{\pi}} \min_{f_s(i,j)} \sum_{i=1}^{N}\sum_{j=1}^{N} M_{ij}\pi(a_j|s)f_s(i, j) \leq \delta$ because if we have a feasible $f_s(i, j)$ to make (10b) hold, we must have $\mathbb{E}_{s \sim \rho_v^{\pi}} \min_{f_s(i,j)} \sum_{i=1}^{N}\sum_{j=1}^{N} M_{ij}\pi(a_j|s)f_s(i, j) \leq \delta$.

Since both the objective function and the constraint are linear in $f_s(i, j)$, (10) is a convex optimization problem. Also, Slater's condition holds for (10) as the feasible region has an interior point, which is $f_s(i, i) = 1 \,\forall i$, and $f_s(i, j) = 0 \,\forall i \neq j$. Meanwhile, since $A^{\pi}(s, a)$ is bounded based on Assumption 1, the objective is bounded above. Therefore, strong duality holds for (10). At this point we can derive the dual problem of (10) as its equivalent reformulation:

$$\min_{\beta \geq 0, \zeta_j^s} \quad \beta\delta + \int_{s \in \mathcal{S}} \sum_{j=1}^{N} \zeta_j^s ds \tag{11}$$

$$s.t. \quad A^{\pi}(s, a_i)\pi(a_j|s) - \beta M_{ij}\pi(a_j|s) - \frac{\zeta_j^s}{\rho_v^{\pi}(s)} \leq 0, \qquad \forall s \in \mathcal{S}, i, j = 1\dots N.$$

We observe that with a fixed $\beta$, the optimal $\zeta_j^s$ will be achieved at:

$$\zeta_j^{s*}(\beta) = \max_{i=1\dots N} \rho_v^{\pi}(s)\pi(a_j|s)(A^{\pi}(s, a_i) - \beta M_{ij}). \tag{12}$$

Denote $\beta^*$ as an optimal solution to (11) and $f_s^*(i,j)$ as an optimal solution to (10). Due to the complimentary slackness, the following equations hold:

$$(A^\pi(s,a_i)\pi(a_j|s) - \beta^* M_{ij}\pi(a_j|s) - \frac{\zeta_j^{s*}(\beta^*)}{\rho_v^\pi(s)})f_s^*(i,j) = 0, \qquad \forall s,i,j.$$

In this case, $f_s^*(i,j)$ can have non-zero values only when $A^\pi(s,a_i)\pi(a_j|s) - \beta^* M_{ij}\pi(a_j|s) - \frac{\zeta_j^{s*}(\beta^*)}{\rho_v^\pi(s)} = 0$, which means $\zeta_j^{s*}(\beta^*) = \rho_v^\pi(s)\pi(a_j|s)(A^\pi(s,a_i) - \beta^* M_{ij})$. Given the expression of the optimal $\zeta_j^{s*}$ in (12), $f_s^*(i,j)$ can have non-zero values only when $i \in \mathcal{K}_s^\pi(\beta^*,j)$, where $\mathcal{K}_s^\pi(\beta,j) = \mathrm{argmax}_{k=1...N} A^\pi(s,a_k) - \beta M_{kj}$. Since $\sum_{i=1}^N f_s^*(i,j) = 1$ as indicated in (10c), we can choose an arbitrary optimizer $k_s^\pi(\beta^*,j) \in \mathcal{K}_s^\pi(\beta^*,j)$ to derive an optimal solution $f_s^*(i,j)$:

$$f_s^*(i,j) = \begin{cases} 1 & \text{if } i = k_s^\pi(\beta^*,j) \\ 0 & \text{otherwise,} \end{cases}$$

And then the corresponding optimal solution is, $\pi^*(a_i|s) = \sum_{j=1}^N \pi(a_j|s)f_s^*(i,j)$.

Last, by substituting $\zeta_j^{s*}(\beta) = \rho_v^\pi(s)\pi(a_j|s)(A^\pi(s,a_{k_s^\pi(\beta,j)}) - \beta M_{k_s^\pi(\beta,j)j})$ into the dual problem (11), we can reformulate (11) into:

$$\min_{\beta \geq 0}\{\beta\delta + \int_{s\in\mathcal{S}}\sum_{j=1}^N \zeta_j^{s*}(\beta)ds\} = \min_{\beta \geq 0}\{\beta\delta + \mathbb{E}_{s\sim\rho_v^\pi}\sum_{j=1}^N \pi(a_j|s)[A^\pi(s,a_{k_s^\pi(\beta,j)}) - \beta M_{k_s^\pi(\beta,j)j}]\}.$$
(13)

The optimal $\beta$ can then be obtained by solving (13).

We will further show that $\beta^* \leq \bar{\beta} := \max_{s\in\mathcal{S},k,j=1...N,k\neq j} (M_{kj})^{-1}(A^\pi(s,a_k) - A^\pi(s,a_j))$.

In the general case, i.e., $\beta \geq 0$, (10a) is non-negative because:

$$\mathbb{E}_{s\sim\rho_v^\pi}\sum_{i=1}^N\sum_{j=1}^N A^\pi(s,a_i)\pi(a_j|s)f_s^*(i,j) \tag{14a}$$

$$= \mathbb{E}_{s\sim\rho_v^\pi}\sum_{j=1}^N \pi(a_j|s)\sum_{i=1}^N A^\pi(s,a_i)f_s^*(i,j) \tag{14b}$$

$$= \mathbb{E}_{s\sim\rho_v^\pi}\sum_{j=1}^N \pi(a_j|s)A^\pi(s,a_{k_s^\pi(\beta^*,j)}) \tag{14c}$$

$$\geq \mathbb{E}_{s\sim\rho_v^\pi}\sum_{j=1}^N \pi(a_j|s)\{A^\pi(s,a_j) + \beta^* M_{k_s^\pi(\beta^*,j)j}\} \tag{14d}$$

$$= \mathbb{E}_{s\sim\rho_v^\pi}\sum_{j=1}^N \pi(a_j|s)\beta^* M_{k_s^\pi(\beta^*,j)j} \tag{14e}$$

$$\geq 0, \tag{14f}$$

where (14d) holds since $A^\pi(s,a_{k_s^\pi(\beta^*,j)}) - \beta^* M_{k_s^\pi(\beta^*,j)j} \geq A^\pi(s,a_j) - \beta^* M_{jj} = A^\pi(s,a_j)$.

When $\beta^* > \max_{s\in\mathcal{S},k,j=1...N,k\neq j}\{\frac{A^\pi(s,a_k)-A^\pi(s,a_j)}{M_{kj}}\}$, we have that for all $s \in \mathcal{S}$, $k_s^\pi(\beta^*,j) = j$. Thus, $f_s^*(i,i) = 1, \forall i$ and $f_s^*(i,j) = 0, \forall i \neq j$. The objective value (10a) will be 0 because $\mathbb{E}_{s\sim\rho_v^\pi}\sum_{i=1}^N\sum_{j=1}^N A^\pi(s,a_i)\pi(a_j|s)f_s^*(i,j) = \mathbb{E}_{s\sim\rho_v^\pi}\sum_{i=1}^N A^\pi(s,a_i)\pi(a_i|s) = 0$. The left hand side of (10b) equals to $\mathbb{E}_{s\sim\rho_v^\pi}\sum_{i=1}^N\sum_{j=1}^N M_{ij}\pi(a_j|s)f_s^*(i,j) = \mathbb{E}_{s\sim\rho_v^\pi}\sum_{i=1}^N M_{ii}\pi(a_i|s) = 0$. Thus, for any $\delta > 0$, (10b) is always satisfied.

Since the objective of the primal Wasserstein trust-region constrained problem in (6) constantly evaluates to 0 when $\beta^* > \max_{s\in\mathcal{S},k,j=1...N,k\neq j}\{\frac{A^\pi(s,a_k)-A^\pi(s,a_j)}{M_{kj}}\}$, and is non-negative when $\beta^* \leq$

$\max_{s \in \mathcal{S}, k,j=1...N, k \neq j} \{ \frac{A^{\pi}(s,a_k) - A^{\pi}(s,a_j)}{M_{kj}} \}$, we can use $\max_{s \in \mathcal{S}, k,j=1...N, k \neq j} \{ \frac{A^{\pi}(s,a_k) - A^{\pi}(s,a_j)}{M_{kj}} \}$ as an upper bound for the optimal dual variable $\beta^*$. $\qquad\square$

## B  PROOF OF THEOREM 2

**Theorem 2.** *(Performance improvement)* *For any initial state distribution $\mu$ and any $\beta_t \geq 0$, if $||\hat{A}^{\pi} - A^{\pi}||_{\infty} \leq \epsilon$ for some $\epsilon > 0$, the WPO policy update with the inaccurate advantage function $\hat{A}^{\pi}$, guarantees the following performance improvement bound,*

$$J(\pi_{t+1}) \geq J(\pi_t) + \beta_t \mathbb{E}_{s \sim \rho_{\mu}^{\pi_{t+1}}} \sum_{j=1}^{N} \pi_t(a_j|s) M_{\hat{k}_s^{\pi_t}(\beta_t,j)j} - \frac{2\epsilon}{1-\gamma}. \tag{7}$$

*Proof of Theorem 2.*

$$J(\pi_{t+1}) - J(\pi_t) = \mathbb{E}_{s \sim \rho_{\mu}^{\pi_{t+1}}} \mathbb{E}_{a \sim \pi_{t+1}}[A^{\pi_t}(s,a)] \tag{15a}$$

$$= \mathbb{E}_{s \sim \rho_{\mu}^{\pi_{t+1}}} \sum_{i=1}^{N} \pi_{t+1}(a_i|s) A^{\pi_t}(s,a_i) \tag{15b}$$

$$= \mathbb{E}_{s \sim \rho_{\mu}^{\pi_{t+1}}} \sum_{i=1}^{N} \sum_{j=1}^{N} \pi_t(a_j|s) \hat{f}_s^t(i,j) A^{\pi_t}(s,a_i) \tag{15c}$$

$$= \mathbb{E}_{s \sim \rho_{\mu}^{\pi_{t+1}}} \sum_{j=1}^{N} \pi_t(a_j|s) \sum_{i=1}^{N} \hat{f}_s^t(i,j) A^{\pi_t}(s,a_i) \tag{15d}$$

$$= \mathbb{E}_{s \sim \rho_{\mu}^{\pi_{t+1}}} \sum_{j=1}^{N} \pi_t(a_j|s) A^{\pi_t}(s, a_{\hat{k}_s^{\pi_t}(\beta_t,j)}) \tag{15e}$$

$$\geq \mathbb{E}_{s \sim \rho_{\mu}^{\pi_{t+1}}} \sum_{j=1}^{N} \pi_t(a_j|s) [A^{\pi_t}(s,a_j) + \beta_t M_{\hat{k}_s^{\pi_t}(\beta_t,j)j} - 2\epsilon] \tag{15f}$$

$$= \beta_t \mathbb{E}_{s \sim \rho_{\mu}^{\pi_{t+1}}} \sum_{j=1}^{N} \pi_t(a_j|s) M_{\hat{k}_s^{\pi_t}(\beta_t,j)j} - \frac{2\epsilon}{1-\gamma}, \tag{15g}$$

where (15a) holds due to the performance difference lemma in Kakade & Langford (2002); (15f) follows from the definition of $\hat{k}_s^{\pi_t}(\beta_t, j)$ and the fact that $||\hat{A}^{\pi_t} - A^{\pi_t}||_{\infty} \leq \epsilon$, therefore $[A^{\pi_t}(s, a_{\hat{k}_s^{\pi_t}(\beta_t,j)}) + \epsilon] - \beta_t M_{\hat{k}_s^{\pi_t}(\beta_t,j)j} \geq \hat{A}^{\pi_t}(s, a_{\hat{k}_s^{\pi_t}(\beta_t,j)}) - \beta_t M_{\hat{k}_s^{\pi_t}(\beta_t,j)j} \geq \hat{A}^{\pi_t}(s,a_j) - \beta_t M_{jj} = \hat{A}^{\pi_t}(s,a_j) \geq A^{\pi_t}(s,a_j) - \epsilon$; (15g) holds since $\mathbb{E}_{a \sim \pi}[A^{\pi}(s,a)] = 0$. $\qquad\square$

## C  PROOF OF THEOREM 3

**Theorem 3.** *If Assumption 1 holds, then the optimal solution to the trust region constrained problem (4) with Sinkhorn divergence is:*

$$\pi_{\lambda}^*(a_i|s) = \sum_{j=1}^{N} \frac{\exp\left(\frac{\lambda}{\beta_{\lambda}^*} A^{\pi}(s,a_i) - \lambda M_{ij}\right)}{\sum_{k=1}^{N} \exp\left(\frac{\lambda}{\beta_{\lambda}^*} A^{\pi}(s,a_k) - \lambda M_{kj}\right)} \pi(a_j|s), \tag{8}$$

*where $M$ denotes the cost matrix and $\beta_{\lambda}^*$ is an optimal solution to the following dual formulation:*

$$\min_{\beta \geq 0} F_{\lambda}(\beta) = \min_{\beta \geq 0} \Big\{ \beta\delta - \mathbb{E}_{s \sim \rho_v^{\pi}} \sum_{j=1}^{N} \pi(a_j|s)(\frac{\beta}{\lambda} + \frac{\beta}{\lambda} \ln(\pi(a_j|s)) -$$

$$\frac{\beta}{\lambda} \ln[\sum_{i=1}^{N} \exp\left(\frac{\lambda}{\beta} A^{\pi}(s,a_i) - \lambda M_{ij}\right)]) + \mathbb{E}_{s \sim \rho_v^{\pi}} \sum_{i=1}^{N} \sum_{j=1}^{N} \frac{\beta}{\lambda} \frac{\exp\left(\frac{\lambda}{\beta} A^{\pi}(s,a_i) - \lambda M_{ij}\right) \cdot \pi(a_j|s)}{\sum_{k=1}^{N} \exp\left(\frac{\lambda}{\beta} A^{\pi}(s,a_k) - \lambda M_{kj}\right)} \Big\}. \tag{9}$$

*Moreover, we have $\beta_{\lambda}^* \leq \frac{2A^{max}}{\delta}$.*

*Proof of Theorem 3.* Invoking the definition of Sinkhorn divergence in (3), the trust region constrained problem with Sinkhorn divergence can be reformulated as:

$$\max_{Q} \quad \mathbb{E}_{s \sim \rho_v^\pi}[\sum_{i=1}^{N} A^\pi(s, a_i) \sum_{j=1}^{N} Q_{ij}^s] \tag{16a}$$

$$s.t. \quad \mathbb{E}_{s \sim \rho_v^\pi}[\sum_{i=1}^{N} \sum_{j=1}^{N} M_{ij} Q_{ij}^s + \frac{1}{\lambda} Q_{ij}^s \log Q_{ij}^s] \leq \delta \tag{16b}$$

$$\sum_{i=1}^{N} Q_{ij}^s = \pi(a_j|s), \quad \forall j = 1, \ldots, N, s \in \mathcal{S}. \tag{16c}$$

Let $\beta$ and $\omega$ represent the dual variables of constraints (16b) and (16c) respectively, then the Lagrangian duality of (16) can be derived as:

$$\max_{Q} \min_{\beta \geq 0, \omega} L(Q, \beta, \omega) = \max_{Q} \min_{\beta \geq 0, \omega} \mathbb{E}_{s \sim \rho_v^\pi}[\sum_{i=1}^{N} A^\pi(s, a_i) \sum_{j=1}^{N} Q_{ij}^s]$$

$$+ \int_{s \in \mathcal{S}} \sum_{j=1}^{N} \omega_j^s (\sum_{i=1}^{N} Q_{ij}^s - \pi(a_j|s)) ds + \beta(\delta - \mathbb{E}_{s \sim \rho_v^\pi}[\sum_{i=1}^{N} \sum_{j=1}^{N} M_{ij} Q_{ij}^s + \frac{1}{\lambda} Q_{ij}^s \log Q_{ij}^s]) \tag{17a}$$

$$= \max_{Q} \min_{\beta \geq 0, \omega} \mathbb{E}_{s \sim \rho_v^\pi}[\sum_{i=1}^{N} A^\pi(s, a_i) \sum_{j=1}^{N} Q_{ij}^s] + \int_{s \in \mathcal{S}} \sum_{j=1}^{N} \sum_{i=1}^{N} \frac{\omega_j^s}{\rho_v^\pi(s)} Q_{ij}^s \rho_v^\pi(s) ds$$

$$- \int_{s \in \mathcal{S}} \sum_{j=1}^{N} \omega_j^s \pi(a_j|s) ds + \beta \delta - \beta \mathbb{E}_{s \sim \rho_v^\pi}[\sum_{i=1}^{N} \sum_{j=1}^{N} M_{ij} Q_{ij}^s + \frac{1}{\lambda} Q_{ij}^s \log Q_{ij}^s]) \tag{17b}$$

$$= \max_{Q} \min_{\beta \geq 0, \omega} \beta \delta - \int_{s \in \mathcal{S}} \sum_{j=1}^{N} \omega_j^s \pi(a_j|s) ds$$

$$+ \mathbb{E}_{s \sim \rho_v^\pi}[\sum_{i=1}^{N} \sum_{j=1}^{N} (A^\pi(s, a_i) - \beta M_{ij} + \frac{\omega_j^s}{\rho_v^\pi(s)}) Q_{ij}^s - \frac{\beta}{\lambda} Q_{ij}^s \log Q_{ij}^s] \tag{17c}$$

$$= \min_{\beta \geq 0, \omega} \max_{Q} \beta \delta - \int_{s \in \mathcal{S}} \sum_{j=1}^{N} \omega_j^s \pi(a_j|s) ds$$

$$+ \mathbb{E}_{s \sim \rho_v^\pi}[\sum_{i=1}^{N} \sum_{j=1}^{N} (A^\pi(s, a_i) - \beta M_{ij} + \frac{\omega_j^s}{\rho_v^\pi(s)}) Q_{ij}^s - \frac{\beta}{\lambda} Q_{ij}^s \log Q_{ij}^s], \tag{17d}$$

where (17d) holds since the Lagrangian function $L(Q, \beta, \omega)$ is concave in $Q$ and linear in $\beta$ and $\omega$, and we can exchange the $\max$ and the $\min$ following the Minimax theorem (Sion, 1958).

Note that the inner max problem of (17d) is an unconstrained concave problem, and we can obtain the optimal $Q$ by taking the derivatives and setting them to 0. That is,

$$\frac{\partial L}{\partial Q_{ij}^s} = A^\pi(s, a_i) - \beta M_{ij} + \frac{\omega_j^s}{\rho_v^\pi(s)} - \frac{\beta}{\lambda}(\log Q_{ij}^s + 1) = 0, \quad \forall i, j = 1, \cdots, N, s \in \mathcal{S}. \tag{18}$$

Therefore, we have the optimal $Q_{ij}^{s*}$ as:

$$Q_{ij}^{s*} = \exp\left(\frac{\lambda}{\beta} A^\pi(s, a_i) - \lambda M_{ij}\right) \exp\left(\frac{\lambda \omega_j^s}{\beta \rho_v^\pi(s)} - 1\right), \quad \forall i, j = 1, \cdots, N, s \in \mathcal{S}. \tag{19}$$

In addition, since $\sum_{i=1}^{N} Q_{ij}^{s*} = \pi(a_j|s)$, we have the following hold:

$$\exp\left(\frac{\lambda \omega_j^s}{\beta \rho_v^\pi(s)} - 1\right) = \frac{\pi(a_j|s)}{\sum_{i=1}^{N} \exp\left(\frac{\lambda}{\beta} A^\pi(s, a_i) - \lambda M_{ij}\right)}. \tag{20}$$

By substituting the left hand side of (20) into (19), we can further reformulate the optimal $Q_{ij}^{s*}$ as:

$$Q_{ij}^{s*} = \frac{\exp\left(\frac{\lambda}{\beta}A^\pi(s,a_i) - \lambda M_{ij}\right)}{\sum_{k=1}^N \exp\left(\frac{\lambda}{\beta}A^\pi(s,a_k) - \lambda M_{kj}\right)}\pi(a_j|s), \quad \forall i,j = 1,\cdots,N, s \in \mathcal{S}. \quad (21)$$

To obtain the optimal dual variables, based on (20), we have the optimal $\omega^*$ as:

$$\omega_j^{s*} = \rho_v^\pi(s)\{\frac{\beta}{\lambda} + \frac{\beta}{\lambda}\ln(\pi(a_j|s)) - \frac{\beta}{\lambda}\ln[\sum_{i=1}^N \exp\left(\frac{\lambda}{\beta}A^\pi(s,a_i) - \lambda M_{ij}\right)]\}, \quad \forall j = 1,\cdots,N, s \in \mathcal{S} \quad (22)$$

By substituting (21) and (22) into (17d), we can obtain the optimal $\beta^*$ via:

$$\min_{\beta \geq 0} \quad \beta\delta - \mathbb{E}_{s\sim\rho_v^\pi}\sum_{j=1}^N \pi(a_j|s)\{\frac{\beta}{\lambda} + \frac{\beta}{\lambda}\ln(\pi(a_j|s)) - \frac{\beta}{\lambda}\ln[\sum_{i=1}^N \exp\left(\frac{\lambda}{\beta}A^\pi(s,a_i) - \lambda M_{ij}\right)]\}$$

$$+ \mathbb{E}_{s\sim\rho_v^\pi}\sum_{i=1}^N\sum_{j=1}^N \frac{\beta}{\lambda}\frac{\exp\left(\frac{\lambda}{\beta}A^\pi(s,a_i) - \lambda M_{ij}\right)\cdot\pi(a_j|s)}{\sum_{k=1}^N \exp\left(\frac{\lambda}{\beta}A^\pi(s,a_k) - \lambda M_{kj}\right)}.$$

The proof for the upper bound of sinkhorn optimal $\beta$ can be found in Appendix D. $\qquad\square$

## D  UPPER BOUND OF SINKHORN OPTIMAL BETA

In this section, we will derive the upper bound of Sinkhorn optimal $\beta$. First, for a given $\beta$, the optimal $Q_{ij}^{s*}(\beta)$ to the Lagrangian dual $L(Q,\beta,\omega)$ can be expressed in (21). With this, we will present the following two lemmas:

**Lemma 1.** *The objective function (16a) with respect to $Q_{ij}^{s*}(\beta)$ decreases as the dual variable $\beta$ increases.*

**Lemma 2.** *If Assumption 1 holds, then for every $\delta > 0$, $Q_{ij}^{s*}(\frac{2A^{max}}{\delta})$ is feasible to (16b) for any $\lambda$.*

We provide proofs for Lemma 1 and Lemma 2 in Appendix D.1 and Appendix D.2 respectively. Given the above two lemmas, we are able to prove the following proposition on the upper bound of Sinkhorn optimal $\beta$:

**Proposition 1.** *If $\beta_\lambda^*$ is the optimal dual solution to the Sinkhorn dual formulation (9), then $\beta_\lambda^* \leq \frac{2A^{max}}{\delta}$ for any $\lambda$.*

*Proof of Proposition 1.* We will prove it by contradiction. According to Lemma 2, $Q_{ij}^{s*}(\frac{2A^{max}}{\delta})$ is feasible to (16b). Since $\beta_\lambda^*$ is the optimal dual solution, $Q_{ij}^{s*}(\beta_\lambda^*)$ is optimal to (16). If $\beta_\lambda^* > \frac{2A^{max}}{\delta}$, according to Lemma 1, the objective value in (16a) with respect to $\frac{2A^{max}}{\delta}$ is smaller than the objective value in (16a) with respect to $\beta_\lambda^*$, which contradicts the fact that $Q_{ij}^{s*}(\beta_\lambda^*)$ is the optimal solution to (16). $\qquad\square$

### D.1  PROOF OF LEMMA 1

**Lemma 1.** *The objective function (16a) with respect to $Q_{ij}^{s*}(\beta)$ decreases as the dual variable $\beta$ increases.*

*Proof of Lemma 1.* Let $G_\lambda(\beta)$ represent the objective function (16a). By substituting the optimal $Q_{ij}^{s*}$ in (21) into (16a), we have:

$$G_\lambda(\beta) = \mathbb{E}_{s\sim\rho_v^\pi}[\sum_{i=1}^N A^\pi(s,a_i)\sum_{j=1}^N \frac{\exp\left(\frac{\lambda}{\beta}A^\pi(s,a_i) - \lambda M_{ij}\right)}{\sum_{k=1}^N \exp\left(\frac{\lambda}{\beta}A^\pi(s,a_k) - \lambda M_{kj}\right)}\pi(a_j|s)] \quad (23a)$$

$$= \mathbb{E}_{s\sim\rho_v^\pi}[\sum_{j=1}^N \pi(a_j|s)\sum_{i=1}^N A^\pi(s,a_i)\frac{\exp\left(\frac{\lambda}{\beta}A^\pi(s,a_i) - \lambda M_{ij}\right)}{\sum_{k=1}^N \exp\left(\frac{\lambda}{\beta}A^\pi(s,a_k) - \lambda M_{kj}\right)}]. \quad (23b)$$

For any $\beta_2 > \beta_1 > 0$, we have:

$$G_\lambda(\beta_1) - G_\lambda(\beta_2)$$

$$= \mathbb{E}_{s\sim\rho_v^\pi} \sum_{j=1}^N \pi(a_j|s) \sum_{i=1}^N A^\pi(s,a_i)\{\frac{\exp\left(\frac{\lambda}{\beta_1}A^\pi(s,a_i) - \lambda M_{ij}\right)}{\sum_{k=1}^N \exp\left(\frac{\lambda}{\beta_1}A^\pi(s,a_k) - \lambda M_{kj}\right)}$$

$$- \frac{\exp\left(\frac{\lambda}{\beta_2}A^\pi(s,a_i) - \lambda M_{ij}\right)}{\sum_{k=1}^N \exp\left(\frac{\lambda}{\beta_2}A^\pi(s,a_k) - \lambda M_{kj}\right)}\} \tag{24a}$$

$$= \mathbb{E}_{s\sim\rho_v^\pi} \sum_{j=1}^N \pi(a_j|s) \sum_{i=1}^N A^\pi(s,a_{[i]})\{\frac{\exp\left(\frac{\lambda}{\beta_1}A^\pi(s,a_{[i]}) - \lambda M_{[i]j}\right)}{\sum_{k=1}^N \exp\left(\frac{\lambda}{\beta_1}A^\pi(s,a_{[k]}) - \lambda M_{[k]j}\right)}$$

$$- \frac{\exp\left(\frac{\lambda}{\beta_2}A^\pi(s,a_{[i]}) - \lambda M_{[i]j}\right)}{\sum_{k=1}^N \exp\left(\frac{\lambda}{\beta_2}A^\pi(s,a_{[k]}) - \lambda M_{[k]j}\right)}\}, \tag{24b}$$

where $[i]$ denotes sorted indices that satisfy $A^\pi(s,a_{[1]}) \geq A^\pi(s,a_{[2]}) \geq \cdots \geq A^\pi(s,a_{[N]})$. Let

$$f_s(i) = \frac{\exp\left(\frac{\lambda}{\beta_1}A^\pi(s,a_{[i]}) - \lambda M_{[i]j}\right)}{\sum_{k=1}^N \exp\left(\frac{\lambda}{\beta_1}A^\pi(s,a_{[k]}) - \lambda M_{[k]j}\right)} - \frac{\exp\left(\frac{\lambda}{\beta_2}A^\pi(s,a_{[i]}) - \lambda M_{[i]j}\right)}{\sum_{k=1}^N \exp\left(\frac{\lambda}{\beta_2}A^\pi(s,a_{[k]}) - \lambda M_{[k]j}\right)} \tag{25a}$$

$$= \frac{\exp\left((\frac{\lambda}{\beta_1} - \frac{\lambda}{\beta_2})A^\pi(s,a_{[i]})\right)\exp\left(\frac{\lambda}{\beta_2}A^\pi(s,a_{[i]}) - \lambda M_{[i]j}\right)}{\sum_{k=1}^N \exp\left((\frac{\lambda}{\beta_1} - \frac{\lambda}{\beta_2})A^\pi(s,a_{[k]})\right)\exp\left(\frac{\lambda}{\beta_2}A^\pi(s,a_{[k]}) - \lambda M_{[k]j}\right)}$$

$$- \frac{\exp\left(\frac{\lambda}{\beta_2}A^\pi(s,a_{[i]}) - \lambda M_{[i]j}\right)}{\sum_{k=1}^N \exp\left(\frac{\lambda}{\beta_2}A^\pi(s,a_{[k]}) - \lambda M_{[k]j}\right)}. \tag{25b}$$

For notation brevity, we let $m_s(i) = \exp\left((\frac{\lambda}{\beta_1} - \frac{\lambda}{\beta_2})A^\pi(s,a_{[i]})\right) > 0$, $w_s(i) = \exp\left(\frac{\lambda}{\beta_2}A^\pi(s,a_{[i]}) - \lambda M_{[i]j}\right) > 0$ and $q_s(i) = \frac{1}{\sum_{k=1}^N m_s(k)w_s(k)} - \frac{1}{\sum_{k=1}^N m_s(i)w_s(k)}$. Then we have

$$(25b) = \frac{m_s(i)w_s(i)}{\sum_{k=1}^N m_s(k)w_s(k)} - \frac{w_s(i)}{\sum_{k=1}^N w_s(k)} \tag{26a}$$

$$= m_s(i)w_s(i)(\frac{1}{\sum_{k=1}^N m_s(k)w_s(k)} - \frac{1}{\sum_{k=1}^N m_s(i)w_s(k)}) \tag{26b}$$

$$= m_s(i)w_s(i)q_s(i). \tag{26c}$$

Since $\frac{\lambda}{\beta_1} - \frac{\lambda}{\beta_2} > 0$, $m_s(i)$ decreases as $i$ increases. Thus, $q_s(i)$ decreases as $i$ increases. Since $m_s(1) \geq m_s(k)$ and $m_s(N) \leq m_s(k)$ for all $k = 1,\ldots,N$, we have $q_s(1) = \frac{1}{\sum_{k=1}^N m_s(k)w_s(k)} - \frac{1}{\sum_{k=1}^N m_s(1)w_s(k)} \geq \frac{1}{\sum_{k=1}^N m_s(k)w_s(k)} - \frac{1}{\sum_{k=1}^N m_s(k)w_s(k)} = 0$, and $q_s(N) = \frac{1}{\sum_{k=1}^N m_s(k)w_s(k)} - \frac{1}{\sum_{k=1}^N m_s(N)w_s(k)} \leq \frac{1}{\sum_{k=1}^N m_s(k)w_s(k)} - \frac{1}{\sum_{k=1}^N m_s(k)w_s(k)} = 0$. Since $q_s(1) \geq 0$, $q_s(N) \leq 0$ and $q_s(i)$ decreases as $i$ increases, there exists an index $1 \leq k_s \leq N$ such that $q_s(i) \geq 0$ for $i \leq k_s$ and $q_s(i) < 0$ for $i > k_s$. Since $m_s(i), w_s(i) > 0$, we have $f_s(i) \geq 0$ for $i \leq k_s$ and $f_s(i) < 0$ for $i > k_s$. In addition, we have $\sum_{i=1}^N f_s(i) = 0$ directly follows from the definition. Thus, $\sum_{i=1}^N f_s(i) = \sum_{i=1}^{k_s} |f_s(i)| - \sum_{i=k_s+1}^N |f_s(i)| = 0$. Therefore,

$$G_\lambda(\beta_1) - G_\lambda(\beta_2) = \mathbb{E}_{s\sim\rho_v^\pi} \sum_{j=1}^N \pi(a_j|s) \sum_{i=1}^N A^\pi(s,a_{[i]})f_s(i) \tag{27a}$$

$$= \mathbb{E}_{s\sim\rho_v^\pi} \sum_{j=1}^N \pi(a_j|s)\{\sum_{i=1}^{k_s} A^\pi(s,a_{[i]})|f_s(i)| - \sum_{i=k_s+1}^N A^\pi(s,a_{[i]})|f_s(i)|\} \tag{27b}$$

$$\geq \mathbb{E}_{s\sim\rho_v^\pi} \sum_{j=1}^N \pi(a_j|s)\{\sum_{i=1}^{k_s} A^\pi(s,a_{[k_s]})|f_s(i)| - \sum_{i=k_s+1}^N A^\pi(s,a_{[k_s+1]})|f_s(i)|\} \tag{27c}$$

$$= \mathbb{E}_{s \sim \rho_v^\pi} \sum_{j=1}^{N} \pi(a_j|s) \{ A^\pi(s, a_{[k_s]}) \sum_{i=1}^{k_s} |f_s(i)| - A^\pi(s, a_{[k_s+1]}) \sum_{i=k_s+1}^{N} |f_s(i)| \} \tag{27d}$$

$$= \mathbb{E}_{s \sim \rho_v^\pi} \sum_{j=1}^{N} \pi(a_j|s) \{ A^\pi(s, a_{[k_s]}) \sum_{i=1}^{k_s} |f_s(i)| - A^\pi(s, a_{[k_s+1]}) \sum_{i=1}^{k_s} |f_s(i)| \} \tag{27e}$$

$$= \mathbb{E}_{s \sim \rho_v^\pi} \sum_{j=1}^{N} \pi(a_j|s) (A^\pi(s, a_{[k_s]}) - A^\pi(s, a_{[k_s+1]})) \sum_{i=1}^{k_s} |f_s(i)| \tag{27f}$$

$$\geq 0. \tag{27g}$$

where (27c) and (27g) hold since $A^\pi(s, a_{[i]})$ is non-increasing as $i$ increases. Furthermore, at least one inequality of (27c) and (27g) will not hold at equality since $\sum_{i=1}^{N} \pi(a_i|s) A^\pi(s, a_i) = 0, \forall s \in \mathcal{S}$, and for non-trivial cases, $Pr\{A^\pi(s, a) = 0, \forall s \in \mathcal{S}, \forall a \in \mathcal{A}\} < 1$, which means $Pr\{\exists s_1, s_2 \in \mathcal{S}, a_1, a_2 \in \mathcal{A}, \ s.t. \ A^\pi(s_1, a_1) \neq A^\pi(s_2, a_2)\} > 0$. Therefore, we have $G_\lambda(\beta_1) - G_\lambda(\beta_2) > 0$. $\quad \square$

### D.2 PROOF OF LEMMA 2

**Lemma 2.** *If Assumption 1 holds, then for every $\delta > 0$, $Q_{ij}^{s*}(\frac{2A^{max}}{\delta})$ is feasible to (16b) for any $\lambda$.*

*Proof of Lemma 2.* By substituting the optimal $Q_{ij}^{s*}$ in (21) into (16b), we can reformulate the left hand side of (16b) as follows:

$$\mathbb{E}_{s \sim \rho_v^\pi} [\sum_{i=1}^{N} \sum_{j=1}^{N} M_{ij} Q_{ij}^{s*} + \frac{1}{\lambda} Q_{ij}^{s*} \log Q_{ij}^{s*}] \tag{28a}$$

$$= \mathbb{E}_{s \sim \rho_v^\pi} \{ \sum_{i=1}^{N} \sum_{j=1}^{N} M_{ij} Q_{ij}^{s*} + \frac{1}{\lambda} Q_{ij}^{s*} [\frac{\lambda}{\beta} A^\pi(s, a_i) - \lambda M_{ij} + \log \frac{\pi(a_j|s)}{\sum_{k=1}^{N} \exp(\frac{\lambda}{\beta} A^\pi(s, a_k) - \lambda M_{kj})}] \} \tag{28b}$$

$$= \mathbb{E}_{s \sim \rho_v^\pi} \{ \sum_{i=1}^{N} \sum_{j=1}^{N} \frac{1}{\beta} Q_{ij}^{s*} A^\pi(s, a_i) + \frac{1}{\lambda} Q_{ij}^{s*} \log \frac{\pi(a_j|s)}{\sum_{k=1}^{N} \exp(\frac{\lambda}{\beta} A^\pi(s, a_k) - \lambda M_{kj})} \}. \tag{28c}$$

Now we prove that when $\beta = \frac{2A^{max}}{\delta}$, $\mathbb{E}_{s \sim \rho_v^\pi} \{ \sum_{i=1}^{N} \sum_{j=1}^{N} \frac{1}{\beta} Q_{ij}^{s*}(\beta) A^\pi(s, a_i) \} \leq \frac{\delta}{2}$ and $\mathbb{E}_{s \sim \rho_v^\pi} \{ \frac{1}{\lambda} Q_{ij}^{s*}(\beta) \log \frac{\pi(a_j|s)}{\sum_{k=1}^{N} \exp(\frac{\lambda}{\beta} A^\pi(s,a_k) - \lambda M_{kj})} \} \leq \frac{\delta}{2}$ hold. For the first part, we have:

$$\mathbb{E}_{s \sim \rho_v^\pi} \{ \sum_{i=1}^{N} \sum_{j=1}^{N} \frac{1}{\beta} Q_{ij}^{s*} A^\pi(s, a_i) \} \tag{29a}$$

$$= \frac{1}{\beta} \mathbb{E}_{s \sim \rho_v^\pi} \{ \sum_{i=1}^{N} [\sum_{j=1}^{N} Q_{ij}^{s*}] A^\pi(s, a_i) \} \tag{29b}$$

$$= \frac{1}{\beta} \mathbb{E}_{s \sim \rho_v^\pi} \{ \sum_{i=1}^{N} \pi'(a_i|s) A^\pi(s, a_i) \} \tag{29c}$$

$$\leq \frac{1}{\beta} \mathbb{E}_{s \sim \rho_v^\pi} \{ \sum_{i=1}^{N} \pi'(a_i|s) |A^\pi(s, a_i)| \} \tag{29d}$$

$$\leq \frac{A^{max}}{\beta} = \frac{\delta}{2}. \tag{29e}$$

For the second part, the followings hold:

$$\mathbb{E}_{s \sim \rho_v^\pi} \{ \sum_{i=1}^{N} \sum_{j=1}^{N} \frac{1}{\lambda} Q_{ij}^{s*} \log \frac{\pi(a_j|s)}{\sum_{k=1}^{N} \exp(\frac{\lambda}{\beta} A^\pi(s, a_k) - \lambda M_{kj})} \} \tag{30a}$$

$$= \mathbb{E}_{s \sim \rho_v^\pi} \{ \sum_{j=1}^N \frac{1}{\lambda} ( \sum_{i=1}^N Q_{ij}^{s*}) \log \frac{\pi(a_j|s)}{\sum_{k=1}^N \exp\left(\frac{\lambda}{\beta} A^\pi(s, a_k) - \lambda M_{kj}\right)} \} \tag{30b}$$

$$= \frac{1}{\lambda} \mathbb{E}_{s \sim \rho_v^\pi} \{ \sum_{j=1}^N \pi(a_j|s) \log \frac{\pi(a_j|s)}{\sum_{k=1}^N \exp\left(\frac{\lambda}{\beta} A^\pi(s, a_k) - \lambda M_{kj}\right)} \} \tag{30c}$$

$$\leq \frac{1}{\lambda} \mathbb{E}_{s \sim \rho_v^\pi} \{ \sum_{j=1}^N \pi(a_j|s) \log \frac{\pi(a_j|s)}{\exp\left(\frac{\lambda}{\beta} A^\pi(s, a_j)\right)} \} \tag{30d}$$

$$\leq \frac{1}{\lambda} \mathbb{E}_{s \sim \rho_v^\pi} \{ \sum_{j=1}^N \pi(a_j|s) \log \frac{1}{\exp\left(\frac{\lambda}{\beta} A^\pi(s, a_j)\right)} \} \tag{30e}$$

$$= \frac{1}{\lambda} \mathbb{E}_{s \sim \rho_v^\pi} \{ \sum_{j=1}^N \pi(a_j|s)(-\frac{\lambda}{\beta} A^\pi(s, a_j)) \} \tag{30f}$$

$$\leq \frac{1}{\beta} \mathbb{E}_{s \sim \rho_v^\pi} \{ \sum_{j=1}^N \pi(a_j|s)|A^\pi(s, a_j)| \} \tag{30g}$$

$$\leq \frac{A^{max}}{\beta} = \frac{\delta}{2}. \tag{30h}$$

Therefore, $Q_{ij}^{s*}(\frac{2A^{\max}}{\delta})$ is feasible to (16b) for any $\lambda$. $\qquad\square$

## E  GRADIENT OF THE OBJECTIVE IN THE SINKHORN DUAL FORMULATION

The closed-form gradient of the objective in the Sinkhorn dual formulation (9) is as follows:

$$\delta - \mathbb{E}_{s \sim \rho_v^\pi} \sum_{j=1}^N \pi(a_j|s) \Big\{ \frac{1}{\lambda} + \frac{1}{\lambda} \ln(\pi(a_j|s)) - \frac{1}{\lambda} \ln[\sum_{i=1}^N \exp\left(\frac{\lambda}{\beta} A^\pi(s, a_i) - \lambda M_{ij}\right)]$$

$$- \frac{\beta}{\lambda} \cdot \frac{1}{\sum_{i=1}^N \exp\left(\frac{\lambda}{\beta} A^\pi(s, a_i) - \lambda M_{ij}\right)} \times \sum_{i=1}^N [\exp\left(\frac{\lambda}{\beta} A^\pi(s, a_i) - \lambda M_{ij}\right) \times -\lambda A^\pi(s, a_i)\beta^{-2}] \Big\}$$

$$+ \mathbb{E}_{s \sim \rho_v^\pi} \sum_{i=1}^N \sum_{j=1}^N \Big\{ \frac{\pi(a_j|s)}{\lambda} \frac{\exp\left(\frac{\lambda}{\beta} A^\pi(s, a_i) - \lambda M_{ij}\right)}{\sum_{k=1}^N \exp\left(\frac{\lambda}{\beta} A^\pi(s, a_k) - \lambda M_{kj}\right)}$$

$$+ \frac{\beta \pi(a_j|s)}{\lambda} \cdot \frac{\exp\left(\frac{\lambda}{\beta} A^\pi(s, a_i) - \lambda M_{ij}\right) \times -\lambda A^\pi(s, a_i)\beta^{-2} \times \sum_{k=1}^N \exp\left(\frac{\lambda}{\beta} A^\pi(s, a_k) - \lambda M_{kj}\right)}{(\sum_{k=1}^N \exp\left(\frac{\lambda}{\beta} A^\pi(s, a_k) - \lambda M_{kj}\right))^2}$$

$$- \frac{\beta \pi(a_j|s)}{\lambda} \cdot \frac{\exp\left(\frac{\lambda}{\beta} A^\pi(s, a_i) - \lambda M_{ij}\right) \times \sum_{k=1}^N [\exp\left(\frac{\lambda}{\beta} A^\pi(s, a_k) - \lambda M_{kj}\right) \times -\lambda A^\pi(s, a_k)\beta^{-2}]}{(\sum_{k=1}^N \exp\left(\frac{\lambda}{\beta} A^\pi(s, a_k) - \lambda M_{kj}\right))^2} \Big\}.$$

## F  PROOF OF THEOREM 4

Given the upper bound of Wassertein optimal $\beta$ in Theorem 1 and the upper bound of Sinkhorn optimal $\beta$ in Proposition 1, we are able to derive the following theorem:

**Theorem 4.** *Define $\beta_{UB} = \max\{\frac{2A^{max}}{\delta}, \bar{\beta}\}$. The following holds:*

1. *$F_\lambda(\beta)$ converges to $F(\beta)$ uniformly on $[0, \beta_{UB}]$,*

2. $\lim_{\lambda \to \infty} \arg\min_{0 \leq \beta \leq \beta_{UB}} F_\lambda(\beta) \subseteq \arg\min_{0 \leq \beta \leq \beta_{UB}} F(\beta)$.

*Proof of Theorem 4.* To show that $F_\lambda(\beta)$ converges to $F(\beta)$ uniformly on $[0, \beta_{\text{UB}}]$, it is equivalent to show that $\lim_{\lambda \to \infty} \sup_{0 \leq \beta \leq \beta_{\text{UB}}} \left| F_\lambda(\beta) - F(\beta) \right| = 0$. Let $\epsilon_s^\pi(\beta, i, j) = A^\pi(s, a_{k_s^\pi(\beta, j)}) -$

$\beta M_{k_s^\pi(\beta,j)j} - [A^\pi(s,a_i) - \beta M_{ij}]$ where $k_s^\pi(\beta,j) \in \mathcal{K}_s^\pi(\beta,j) = \text{argmax}_{k=1\ldots N} A^\pi(s,a_k) - \beta M_{kj}$ is an arbitrary optimizer, and $\epsilon_s^\pi(\beta,i,j) \geq 0$. First, we have

$$\left| F_\lambda(\beta) - F(\beta) \right|$$

$$= \left| \beta\delta - \mathbb{E}_{s\sim\rho_v^\pi} \sum_{j=1}^N \pi(a_j|s)\{\frac{\beta}{\lambda} + \frac{\beta}{\lambda}\ln(\pi(a_j|s)) - \frac{\beta}{\lambda}\ln[\sum_{i=1}^N \exp\left(\frac{\lambda}{\beta}A^\pi(s,a_i) - \lambda M_{ij}\right)]\} \right.$$

$$+ \mathbb{E}_{s\sim\rho_v^\pi} \sum_{i=1}^N \sum_{j=1}^N \frac{\beta}{\lambda} \frac{\exp\left(\frac{\lambda}{\beta}A^\pi(s,a_i) - \lambda M_{ij}\right)\cdot\pi(a_j|s)}{\sum_{k=1}^N \exp\left(\frac{\lambda}{\beta}A^\pi(s,a_k) - \lambda M_{kj}\right)} - \beta\delta$$

$$\left. - \mathbb{E}_{s\sim\rho_v^\pi} \sum_{j=1}^N \pi(a_j|s)[A^\pi(s,a_{k_s^\pi(\beta,j)}) - \beta M_{k_s^\pi(\beta,j)j}] \right| \tag{31a}$$

$$\leq \left| \frac{\beta}{\lambda}\mathbb{E}_{s\sim\rho_v^\pi} \sum_{j=1}^N \pi(a_j|s) \right| + \left| \frac{\beta}{\lambda}\mathbb{E}_{s\sim\rho_v^\pi} \sum_{j=1}^N \pi(a_j|s)\ln(\pi(a_j|s)) \right|$$

$$+ \left| \mathbb{E}_{s\sim\rho_v^\pi} \sum_{i=1}^N \sum_{j=1}^N \frac{\beta}{\lambda} \frac{\exp\left(\frac{\lambda}{\beta}A^\pi(s,a_i) - \lambda M_{ij}\right)\cdot\pi(a_j|s)}{\sum_{k=1}^N \exp\left(\frac{\lambda}{\beta}A^\pi(s,a_k) - \lambda M_{kj}\right)} \right|$$

$$+ \left| \mathbb{E}_{s\sim\rho_v^\pi} \sum_{j=1}^N \pi(a_j|s)\frac{\beta}{\lambda}\ln[\sum_{i=1}^N \exp\left(\frac{\lambda}{\beta}A^\pi(s,a_i) - \lambda M_{ij}\right)] \right.$$

$$\left. - \mathbb{E}_{s\sim\rho_v^\pi} \sum_{j=1}^N \pi(a_j|s)[A^\pi(s,a_{k_s^\pi(\beta,j)}) - \beta M_{k_s^\pi(\beta,j)j}] \right| \tag{31b}$$

$$\leq 2\left| \frac{\beta}{\lambda}\mathbb{E}_{s\sim\rho_v^\pi} \sum_{j=1}^N \pi(a_j|s) \right| + \left| \frac{\beta}{\lambda}\mathbb{E}_{s\sim\rho_v^\pi} \sum_{j=1}^N \pi(a_j|s)\ln(\pi(a_j|s)) \right|$$

$$+ \left| \mathbb{E}_{s\sim\rho_v^\pi} \sum_{j=1}^N \pi(a_j|s)\frac{\beta}{\lambda}\ln[\sum_{i=1}^N \exp\left(\frac{\lambda}{\beta}A^\pi(s,a_i) - \lambda M_{ij}\right)] \right.$$

$$\left. - \mathbb{E}_{s\sim\rho_v^\pi} \sum_{j=1}^N \pi(a_j|s)[A^\pi(s,a_{k_s^\pi(\beta,j)}) - \beta M_{k_s^\pi(\beta,j)j}] \right|. \tag{31c}$$

In addition,

$$\left| \mathbb{E}_{s\sim\rho_v^\pi} \sum_{j=1}^N \pi(a_j|s)\frac{\beta}{\lambda}\ln[\sum_{i=1}^N \exp\left(\frac{\lambda}{\beta}A^\pi(s,a_i) - \lambda M_{ij}\right)] \right.$$

$$\left. - \mathbb{E}_{s\sim\rho_v^\pi} \sum_{j=1}^N \pi(a_j|s)[A^\pi(s,a_{k_s^\pi(\beta,j)}) - \beta M_{k_s^\pi(\beta,j)j}] \right| \tag{32a}$$

$$= \left| \mathbb{E}_{s\sim\rho_v^\pi} \sum_{j=1}^N \pi(a_j|s)\frac{\beta}{\lambda}\ln[\exp\left(\frac{\lambda}{\beta}A^\pi(s,a_{k_s^\pi(\beta,j)}) - \lambda M_{k_s^\pi(\beta,j)j}\right)\sum_{i=1}^N \exp\left(-\frac{\lambda}{\beta}\epsilon_s^\pi(\beta,i,j)\right)] \right.$$

$$\left. - \mathbb{E}_{s\sim\rho_v^\pi} \sum_{j=1}^N \pi(a_j|s)[A^\pi(s,a_{k_s^\pi(\beta,j)}) - \beta M_{k_s^\pi(\beta,j)j}] \right| \tag{32b}$$

$$= \left| \mathbb{E}_{s\sim\rho_v^\pi} \sum_{j=1}^N \pi(a_j|s)\frac{\beta}{\lambda}\{\ln[\exp\left(\frac{\lambda}{\beta}A^\pi(s,a_{k_s^\pi(\beta,j)}) - \lambda M_{k_s^\pi(\beta,j)j}\right)] + \ln[\sum_{i=1}^N \exp\left(-\frac{\lambda}{\beta}\epsilon_s^\pi(\beta,i,j)\right)]\} \right.$$

$$\left. - \mathbb{E}_{s\sim\rho_v^\pi} \sum_{j=1}^N \pi(a_j|s)[A^\pi(s,a_{k_s^\pi(\beta,j)}) - \beta M_{k_s^\pi(\beta,j)j}] \right| \tag{32c}$$

$$= \left| \mathbb{E}_{s \sim \rho_v^\pi} \sum_{j=1}^N \pi(a_j|s) \frac{\beta}{\lambda} \ln[\sum_{i=1}^N \exp\left(-\frac{\lambda}{\beta}\epsilon_s^\pi(\beta,i,j)\right)] \right|. \tag{32d}$$

Therefore,

$$\lim_{\lambda \to \infty} \sup_{0 \le \beta \le \beta_{\text{UB}}} \left| F_\lambda(\beta) - F(\beta) \right| \tag{33a}$$

$$\le \lim_{\lambda \to \infty} \frac{2\beta_{\text{UB}}}{\lambda} \left| \mathbb{E}_{s \sim \rho_v^\pi} \sum_{j=1}^N \pi(a_j|s) \right| + \lim_{\lambda \to \infty} \frac{\beta_{\text{UB}}}{\lambda} \left| \mathbb{E}_{s \sim \rho_v^\pi} \sum_{j=1}^N \pi(a_j|s) \ln(\pi(a_j|s)) \right|$$

$$+ \lim_{\lambda \to \infty} \sup_{0 \le \beta \le \beta_{\text{UB}}} \frac{\beta}{\lambda} \left| \mathbb{E}_{s \sim \rho_v^\pi} \sum_{j=1}^N \pi(a_j|s) \ln[\sum_{i=1}^N \exp\left(-\frac{\lambda}{\beta}\epsilon_s^\pi(\beta,i,j)\right)] \right| \tag{33b}$$

$$= \lim_{\lambda \to \infty} \sup_{0 \le \beta \le \beta_{\text{UB}}} \frac{\beta}{\lambda} \left| \mathbb{E}_{s \sim \rho_v^\pi} \sum_{j=1}^N \pi(a_j|s) \ln[\sum_{i=1}^N \exp\left(-\frac{\lambda}{\beta}\epsilon_s^\pi(\beta,i,j)\right)] \right|. \tag{33c}$$

In addition, $\forall \beta \in [0, \beta_{\text{UB}}]$ and $\forall \lambda$, $\epsilon_s^\pi(\beta,i,j)$ is bounded since

$$\left| \epsilon_s^\pi(\beta,i,j) \right| = \left| A^\pi(s, a_{k_s^\pi(\beta,j)}) - \beta M_{k_s^\pi(\beta,j)j} - [A^\pi(s,a_i) - \beta M_{ij}] \right| \tag{34}$$

$$= \left| (A^\pi(s, a_{k_s^\pi(\beta,j)}) - A^\pi(s,a_i)) - (\beta M_{k_s^\pi(\beta,j)j} - \beta M_{ij}) \right|$$

$$\le \left| A^\pi(s, a_{k_s^\pi(\beta,j)}) - A^\pi(s,a_i) \right| + \left| \beta M_{k_s^\pi(\beta,j)j} - \beta M_{ij} \right|$$

$$\le 2 \max_{s,a} A^\pi(s,a) + \beta_{\text{UB}} \max_{i,j} M_{ij}$$

$$\le 2A^{\max} + \beta_{\text{UB}} \max_{i,j} M_{ij} < \infty. \tag{35}$$

Then, $\left| \mathbb{E}_{s \sim \rho_v^\pi} \sum_{j=1}^N \pi(a_j|s) \ln[\sum_{i=1}^N \exp\left(-\frac{\lambda}{\beta}\epsilon_s^\pi(\beta,i,j)\right)] \right|$ is bounded. Therefore in (33c), the optimal $\beta$ can be achieved. Let $\beta^\lambda = \arg\max_{0 \le \beta \le \beta_{\text{UB}}} \frac{\beta}{\lambda} \left| \mathbb{E}_{s \sim \rho_v^\pi} \sum_{j=1}^N \pi(a_j|s) \ln[\sum_{i=1}^N \exp\left(-\frac{\lambda}{\beta}\epsilon_s^\pi(\beta,i,j)\right)] \right|$, and then we have:

$$\lim_{\lambda \to \infty} \sup_{0 \le \beta \le \beta_{\text{UB}}} \frac{\beta}{\lambda} \left| \mathbb{E}_{s \sim \rho_v^\pi} \sum_{j=1}^N \pi(a_j|s) \ln[\sum_{i=1}^N \exp\left(-\frac{\lambda}{\beta}\epsilon_s^\pi(\beta,i,j)\right)] \right| \tag{36a}$$

$$= \lim_{\lambda \to \infty} \frac{\beta^\lambda}{\lambda} \left| \mathbb{E}_{s \sim \rho_v^\pi} \sum_{j=1}^N \pi(a_j|s) \ln[\sum_{i=1}^N \exp\left(-\frac{\lambda}{\beta^\lambda}\epsilon_s^\pi(\beta^\lambda,i,j)\right)] \right|. \tag{36b}$$

Define $\sigma_s(j) = \min_{0 \le \beta \le \beta_{\text{UB}}} \min_{i=1 \dots N, i \notin \mathcal{K}_s^\pi(\beta,j)} \epsilon_s^\pi(\beta,i,j)$. Then since $\epsilon_s^\pi(\beta,i,j) > 0$ for $i \notin \mathcal{K}_s^\pi(\beta,j)$ based on its definition, we have $\sigma_s(j) > 0$. On one hand, we have

$$\lim_{\lambda \to \infty} \ln[\sum_{i=1}^N \exp\left(-\frac{\lambda}{\beta^\lambda}\epsilon_s^\pi(\beta^\lambda,i,j)\right)] \tag{37a}$$

$$= \lim_{\lambda \to \infty} \ln[\sum_{i=1 | i \notin \mathcal{K}_s^\pi(\beta_\lambda,j)}^N \exp\left(-\frac{\lambda}{\beta^\lambda}\epsilon_s^\pi(\beta^\lambda,i,j)\right) + \sum_{i=1 | i \in \mathcal{K}_s^\pi(\beta_\lambda,j)}^N \exp\left(-\frac{\lambda}{\beta^\lambda}\epsilon_s^\pi(\beta^\lambda,i,j)\right)] \tag{37b}$$

$$\le \lim_{\lambda \to \infty} \ln[\sum_{i=1 | i \notin \mathcal{K}_s^\pi(\beta_\lambda,j)}^N \exp\left(-\frac{\lambda}{\beta_{\text{UB}}}\sigma_s(j)\right) + \sum_{i=1 | i \in \mathcal{K}_s^\pi(\beta_\lambda,j)}^N \exp\left(0\right)] \tag{37c}$$

$$= \lim_{\lambda \to \infty} \ln[\sum_{i=1 | i \notin \mathcal{K}_s^\pi(\beta_\lambda,j)}^N \exp\left(-\frac{\lambda}{\beta_{\text{UB}}}\sigma_s(j)\right) + |\mathcal{K}_s^\pi(\beta_\lambda,j)|] \tag{37d}$$

$$= \lim_{\lambda \to \infty} \ln[|\mathcal{K}_s^\pi(\beta_\lambda,j)|]. \tag{37e}$$

On the other hand, we have

$$\lim_{\lambda\to\infty} \ln[\sum_{i=1}^{N} \exp\left(-\frac{\lambda}{\beta^\lambda}\epsilon_s^\pi(\beta^\lambda, i, j)\right)] \tag{38a}$$

$$= \lim_{\lambda\to\infty} \ln[\sum_{i=1|i\notin\mathcal{K}_s^\pi(\beta_\lambda, j)}^{N} \exp\left(-\frac{\lambda}{\beta^\lambda}\epsilon_s^\pi(\beta^\lambda, i, j)\right) + \sum_{i=1|i\in\mathcal{K}_s^\pi(\beta_\lambda, j)}^{N} \exp\left(-\frac{\lambda}{\beta^\lambda}\epsilon_s^\pi(\beta^\lambda, i, j)\right)] \tag{38b}$$

$$\geq \lim_{\lambda\to\infty} \ln[\sum_{i=1|i\in\mathcal{K}_s^\pi(\beta_\lambda, j)}^{N} \exp\left(-\frac{\lambda}{\beta^\lambda}\epsilon_s^\pi(\beta^\lambda, i, j)\right)] \tag{38c}$$

$$= \lim_{\lambda\to\infty} \ln[\sum_{i=1|i\in\mathcal{K}_s^\pi(\beta_\lambda, j)}^{N} \exp(0)] \tag{38d}$$

$$= \lim_{\lambda\to\infty} \ln[|\mathcal{K}_s^\pi(\beta_\lambda, j)|]. \tag{38e}$$

Therefore, $\lim_{\lambda\to\infty}\left|\ln[\sum_{i=1}^{N}\exp\left(-\frac{\lambda}{\beta^\lambda}\epsilon_s^\pi(\beta^\lambda, i, j)\right)]\right| = \lim_{\lambda\to\infty}\ln[|\mathcal{K}_s^\pi(\beta_\lambda, j)|]$. Based on that, we have

$$\lim_{\lambda\to\infty} \frac{\beta^\lambda}{\lambda}\left|\mathbb{E}_{s\sim\rho_v^\pi}\sum_{j=1}^{N}\pi(a_j|s)\ln[\sum_{i=1}^{N}\exp\left(-\frac{\lambda}{\beta^\lambda}\epsilon_s^\pi(\beta^\lambda, i, j)\right)]\right| \tag{39a}$$

$$\leq \lim_{\lambda\to\infty} \frac{\beta^\lambda}{\lambda}\left|\sum_{j=1}^{N}\ln[\sum_{i=1}^{N}\exp\left(-\frac{\lambda}{\beta^\lambda}\epsilon_s^\pi(\beta^\lambda, i, j)\right)]\right| \tag{39b}$$

$$\leq \lim_{\lambda\to\infty} \frac{\beta^\lambda}{\lambda}\sum_{j=1}^{N}\left|\ln[\sum_{i=1}^{N}\exp\left(-\frac{\lambda}{\beta^\lambda}\epsilon_s^\pi(\beta^\lambda, i, j)\right)]\right| \tag{39c}$$

$$= \lim_{\lambda\to\infty} \frac{\beta^\lambda}{\lambda}\sum_{j=1}^{N}\ln[|\mathcal{K}_s^\pi(\beta_\lambda, j)|] \tag{39d}$$

$$\leq \lim_{\lambda\to\infty} \frac{\beta_{\text{UB}}}{\lambda} N\ln N = 0, \tag{39e}$$

which means $\lim_{\lambda\to\infty}\sup_{0\leq\beta\leq\beta_{\text{UB}}}\left|F_\lambda(\beta) - F(\beta)\right| \leq 0$. Furthermore, since $\lim_{\lambda\to\infty}\sup_{0\leq\beta\leq\beta_{\text{UB}}}|F_\lambda(\beta)-F(\beta)| \geq 0$ holds naturally, we have $\lim_{\lambda\to\infty}\sup_{0\leq\beta\leq\beta_{\text{UB}}}|F_\lambda(\beta)-F(\beta)| = 0$. Therefore, $F_\lambda(\beta)$ converges to $F(\beta)$ uniformly on $[0, \beta_{\text{UB}}]$, which also indicates $F_\lambda(\beta)$ epi-converges to $F(\beta)$ on $[0, \beta_{\text{UB}}]$ (Royset, 2018; Rockafellar & Wets, 1998). By properties of epi-convergence, we have that $\lim_{\lambda\to\infty}\arg\min_{0\leq\beta\leq\beta_{\text{UB}}}F_\lambda(\beta) \subseteq \arg\min_{0\leq\beta\leq\beta_{\text{UB}}}F(\beta)$ (Rockafellar & Wets, 1998). $\square$

## G  OPTIMAL BETA FOR A SPECIAL DISTANCE

**Proposition 2.** *(1). If the initial point $\beta_0$ is in $[\max_{s,j}\{A^\pi(s, a_{k_s}) - A^\pi(s, a_j)\}, +\infty)$, the local optimal $\beta$ solution is $\max_{s,j}\{A^\pi(s, a_{k_s}) - A^\pi(s, a_j)\}$.*
*(2). If the initial point $\beta_0$ is in $[0, \min_{s,j\neq k_s}\{A^\pi(s, a_{k_s}) - A^\pi(s, a_j)\}]$: if $\delta - \int_{s\in\mathcal{S}}\rho^\pi(s)(1 - \pi(a_{k_s}|s))ds < 0$, the local optimal $\beta$ is $\min_{s,j\neq k_s}\{A^\pi(s, a_{k_s}) - A^\pi(s, a_j)\}$; otherwise, the local optimal $\beta$ solution is $0$.*

*(3). If the initial point $\beta_0$ is in $(\min_{s,j\neq k_s}\{A^\pi(s, a_{k_s}) - A^\pi(s, a_j)\}, \max_{s,j}\{A^\pi(s, a_{k_s}) - A^\pi(s, a_j)\})$, we construct sets $I_s^1$ and $I_s^2$ as:*

**for** $s \in \mathcal{S}, j \in \{1, 2 \ldots N\}$ : **if** $\beta_0 \geq A^\pi(s, a_{k_s}) - A^\pi(s, a_j)$ **then** Add $j$ to $I_s^1$ **else** Add $j$ to $I_s^2$. *Then, if $\delta - \mathbb{E}_{s\sim\rho^\pi}\sum_{j\in I_s^2}\pi(a_j|s) < 0$, the local optimal $\beta$ is $\min_{s\in\mathcal{S}, j\in I_s^2}\{A^\pi(s, a_{k_s}) - A^\pi(s, a_j)\}$; otherwise, the local optimal $\beta$ is $\max_{s\in\mathcal{S}, j\in I_s^1}\{A^\pi(s, a_{k_s}) - A^\pi(s, a_j)\}$.*

*Proof of Proposition 2.* (1). When $\beta \in [\max_{s,j}\{A^\pi(s, a_{k_s}) - A^\pi(s, a_j)\}, +\infty)$, we have $A^\pi(s, a_j) \geq A^\pi(s, a_{k_s}) - \beta$ for all $s \in \mathcal{S}$, $j = 1 \ldots N$. Since $A^\pi(s, a_{k_s}) - \beta \geq A^\pi(s, a_k) - \beta$ for all $k = 1 \ldots N$, we have $A^\pi(s, a_j) \geq A^\pi(s, a_k) - \beta$ for all $s \in \mathcal{S}$, $j = 1 \ldots N$, $k = 1 \ldots N$. Thus, $j \in \mathcal{K}_s^\pi(\beta^*, j)$, for all $s \in \mathcal{S}$, $j = 1 \ldots N$. Therefore, (6) can be reformulated as:

$$\min_{\beta \geq 0}\{\beta\delta + \mathbb{E}_{s\sim\rho_v^\pi}\sum_{j=1}^N \pi(a_j|s)A^\pi(s, a_j)\}.$$

Since $\delta \geq 0$, we have the local optimal $\beta = \max_{s,j}\{A^\pi(s, a_{k_s}) - A^\pi(s, a_j)\}$.

(2). When $\beta \in [0, \min_{s,j\neq k_s}\{A^\pi(s, a_{k_s}) - A^\pi(s, a_j)\}]$, we have $A^\pi(s, a_j) \leq A^\pi(s, a_{k_s}) - \beta$ for all $s \in \mathcal{S}$, $j = 1 \ldots N$, $j \neq k_s$. Thus $k_s \in \mathcal{K}_s^\pi(\beta^*, j)$ for all $s \in \mathcal{S}$, $j = 1 \ldots N$. The inner part of (6) then is:

$$\beta\delta + \mathbb{E}_{s\sim\rho_v^\pi}\{\sum_{j=1,j\neq k_s}^N \pi(a_j|s)(A^\pi(s, a_{k_s}) - \beta) + \pi(a_{k_s}|s)A^\pi(s, a_{k_s})\}$$

$$= \beta(\delta - \mathbb{E}_{s\sim\rho_v^\pi}\sum_{j=1,j\neq k_s}^N \pi(a_j|s)) + \mathbb{E}_{s\sim\rho_v^\pi}\sum_{j=1}^N \pi(a_j|s)A^\pi(s, a_{k_s})$$

$$= \beta(\delta - \int_{s\in\mathcal{S}} \rho_v^\pi(s)(1 - \pi(a_{k_s}|s))ds) + \mathbb{E}_{s\sim\rho_v^\pi}\sum_{j=1}^N \pi(a_j|s)A^\pi(s, a_{k_s}).$$

If $\delta - \int_{s\in\mathcal{S}} \rho_v^\pi(s)(1 - \pi(a_{k_s}|s))ds < 0$, we have the local optimal $\beta = \min_{s,j\neq k_s}\{A^\pi(s, a_{k_s}) - A^\pi(s, a_j)\}$. If $\delta - \int_{s\in\mathcal{S}} \rho_v^\pi(s)(1 - \pi(a_{k_s}|s))ds \geq 0$, we have the local optimal $\beta = 0$.

(3). For an initial point $\beta_0$ in $(\min_{s,j\neq k_s}\{A^\pi(s, a_{k_s}) - A^\pi(s, a_j)\}, \max_{s,j}\{A^\pi(s, a_{k_s}) - A^\pi(s, a_j)\})$, we construct partitions $I_s^1$ and $I_s^2$ of the set $\{1, 2 \ldots N\}$ in the way described in Proposition 2 for all $s \in \mathcal{S}$. Consider $\beta$ in the neighborhood of $\beta_0$, i.e., $\beta \geq A^\pi(s, a_{k_s}) - A^\pi(s, a_j)$ for $s \in \mathcal{S}, j \in I_s^1$ and $\beta \leq A^\pi(s, a_{k_s}) - A^\pi(s, a_j)$ for $s \in \mathcal{S}, j \in I_s^2$. Then the inner part of (6) can be reformulated as:

$$\beta\delta + \mathbb{E}_{s\sim\rho_v^\pi}\{\sum_{j\in I_s^1} \pi(a_j|s)A^\pi(s, a_j) + \sum_{j\in I_s^2} \pi(a_j|s)(A^\pi(s, a_{k_s}) - \beta)\}$$

$$= \beta(\delta - \mathbb{E}_{s\sim\rho_v^\pi}\sum_{j\in I_s^2} \pi(a_j|s)) + \mathbb{E}_{s\sim\rho_v^\pi}\{\sum_{j\in I_s^1} \pi(a_j|s)A^\pi(s, a_j) + \sum_{j\in I_s^2} \pi(a_j|s)A^\pi(s, a_{k_s})\}.$$

If $\delta - \mathbb{E}_{s\sim\rho_v^\pi}\sum_{j\in I_s^2} \pi(a_j|s) < 0$, we have the local optimal $\beta = \min_{s\in\mathcal{S}, j\in I_s^2}\{A^\pi(s, a_{k_s}) - A^\pi(s, a_j)\}$. If $\delta - \mathbb{E}_{s\sim\rho_v^\pi}\sum_{j\in I_s^2} \pi(a_j|s) \geq 0$, we have the local optimal $\beta = \max_{s\in\mathcal{S}, j\in I_s^1}\{A^\pi(s, a_{k_s}) - A^\pi(s, a_j)\}$. $\qquad\square$

## H IMPLEMENTATION DETAILS

**Visitation Frequencies Estimation:** The unnormalized discounted visitation frequencies are needed to compute the global optimal $\beta^*$. At the $k$-th iteration, the visitation frequencies $\rho_k^\pi$ are estimated using samples of the trajectory set $\mathcal{D}_k$. Specifically, we first initialize $\rho_k^\pi(s) = 0$, $\forall s \in S$. Then for each timestep $t$ in each trajectory from $\mathcal{D}_k$, we update $\rho_k^\pi$ as $\rho_k^\pi(s_t) \leftarrow \rho_k^\pi(s_t) + \gamma^t/|\mathcal{D}_k|$.

**Policy Representation:** The general approach depicted in Algorithm 1 allows various policy representations including arrays and neural networks. Let $\mathcal{S}_k \subseteq \mathcal{S}$ represent a subset of states to perform the policy update at the $k$-th iteration. When an array is used, the policy update step is simply $\pi_{k+1}(\cdot|s) = \mathbb{F}(\pi_k)(\cdot|s)$, $\forall s \in \mathcal{S}_k$. When a neural network is employed, the policy update step can be achieved by obtaining the gradient descent, i.e., $\nabla \sum_{s\in\mathcal{S}_k}(\mathbb{F}(\pi_k)(\cdot|s) - \pi_k(\cdot|s))^2$.

**Policy Updating Strategy:**

- State space: For environments with a discrete state space (e.g., tabular domains), the WPO/SPO policy update is performed for all states at each iteration. For the environments

with a continuous state space, a random subset of states is sampled at each iteration to perform the policy update.

- Action space: For environments with a continuous action space, we first discretize the action space following Tang & Agrawal (2020) and then WPO/SPO policy update is performed on the discretized action space.

**Hyperparamaters and Additional Results:**

Our main experimental results are reported in Section 6. In addition, we provide the setting of hyperparamaters and network sizes of our WPO/SPO algorithms in Table 3. And we present the numerical results of the final performance comparison among our algorithms and the baseline methods (i.e., TRPO, PPO, A2C) in Table 4.

Table 3: Hyperparamaters and network sizes

|  | Taxi-v3, NChain-v0 CliffWalking-v0 | CartPole-v1 | Acrobot-v1 |
|---|---|---|---|
| $\gamma$ | 0.9 | 0.95 | 0.95 |
| $lr_\pi$ | \ | $10^{-2}$ | $5 \times 10^{-3}$ |
| $lr_{\text{value}}$ | $10^{-2}$ | $10^{-2}$ | $5 \times 10^{-3}$ |
| $|\mathcal{D}_k|$ | 60 (Taxi); 1 (Chain); 3 (CliffWalking) | 2 | 3 |
| $\pi$ size | 2D array | $[64, 64]$ | $[64, 64]$ |
| Q/v size | $[10, 7, 5]$ | $[64, 64]$ | $[64, 64]$ |
| $\lambda$ | $5, 50, 10$ | 10 | 10 |

Table 4: Averaged rewards over last 10% episodes during the training process

| Environment | WPO | SPO | TRPO | PPO | A2C |
|---|---|---|---|---|---|
| Taxi-v3 | $-45 \pm 27$ | $-87 \pm 11$ | $-202 \pm 3$ | $-381 \pm 34$ | $-338 \pm 30$ |
| NChain-v0 | $3549 \pm 197$ | $3432 \pm 131$ | $3522 \pm 258$ | $3506 \pm 237$ | $1606 \pm 10$ |
| CliffWalking-v0 | $-35 \pm 15$ | $-25 \pm 1$ | $-159 \pm 94$ | $-3290 \pm 2106$ | $-5587 \pm 1942$ |
| CartPole-v1 | $388 \pm 54$ | $370 \pm 30$ | $297 \pm 65$ | $193 \pm 45$ | $267 \pm 61$ |
| Acrobot-v1 | $-162 \pm 8$ | $-185 \pm 15$ | $-248 \pm 33$ | $-103 \pm 5$ | $-379 \pm 39$ |

**Additional Experiments on Ablation Study:**

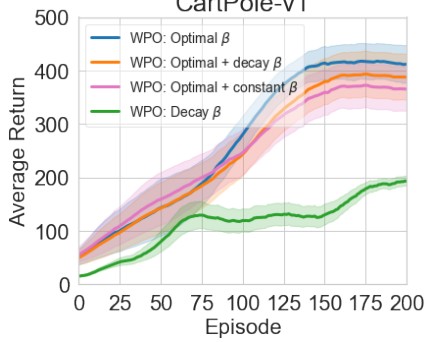 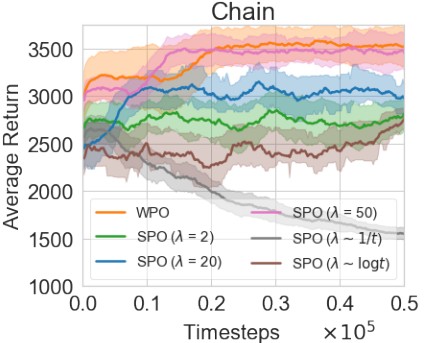

Figure 7: Episode rewards during the training process for different $\beta$ and $\lambda$ settings, averaged across 3 runs with a random initialization. The shaded area depicts the mean $\pm$ the standard deviation.

