# OpenReview forum: "Efficient Wasserstein and Sinkhorn Policy Optimization"
_ICLR.cc/2022/Conference — ICLR 2022 Submitted_

### Official Review · Reviewer_2JvQ · 2021-11-02

**Correctness:** 4
**Technical Novelty And Significance:** 2
**Empirical Novelty And Significance:** 3
**Recommendation:** 6
**Confidence:** 4

**Main Review:**

Review of EFFICIENT WASSERSTEIN AND SINKHORN POLICY OPTIMIZATION

In general, I found the paper to be a nice, if modest, extension of TRPO. In essence, the key idea is trading KL regularization for Wasserstein or Sinkhorn. Experiments show this is, in general, a win, particularly for the case of approximated advantage functions. I found the exposition to be mostly fine with some minor critiques. First, it would be better to have a more detailed treatment of the prior work (TRPO is properly introduced later). Second, separately discussing Wasserstein and Sinkhorn is unnecessary and redundant. They are very closely related and the text (and math) would benefit from making that clearer.


Additional points:
- The discussion of related work is inadequate.
- It would be desirable to make a clear statement of what the reasons for and consequences of assumption 1 are.
- "Compared to the performance bound when using KL-based trust region
(see, e.g., Schulman et al. (2015); Cen et al. (2020)), using the Wasserstein metric yields a tighter performance improvement bound and is more robust to the choice of parameters βt." It would be nice to show this explicitly.
- Theorems 1 and 3 are closely related, as are Wasserstein and Sinkhorn. Is is really necessary to break them into separate sections? This choice makes the relation less clear.

**Summary Of The Paper:**

The paper introduces Wasserstein and Sinkhorn policy optimization, with three main contributions. First, the paper derives closed form expressions for updates using Lagrange multipliers. Second, it proves monotonicity of performance improvement. Third, experiments show efficiency and effectiveness relative to TRPO and PPO.


**Summary Of The Review:**

Overall, this is not the most ambitious paper, but generally a nice contribution.

---

> ### Author Response · Authors · 2021-11-22
> **Response to Reviewer 2JvQ**
>
> Thank you for your valuable comments and for appreciating our work. We have adopted most of the suggestions in our updated submission.  We added a clear statement for Assumption 1 and an explicit performance bound for KL-based trust region. We also provided more discussions on prior work on TRPO and related work using Wasserstein and Sinkhorn distances.
>
> **Q1. Suggestion on the exposition of WPO and SPO.**
>
> **Answer:**
> We fully agree with the reviewer that these two results are not standalone, but rather closely related since Sinkhorn PO is an efficient approximation of Wasserstein PO. On the other hand, we feel it may not be appropriate to present Sinkhorn PO simply as a subsidiary of Wasserstein PO. Our numerical experiments show that SPO sometimes could outperform WPO. We will re-organize the sections and try to make the relation more clear in the final revision.

---

> > ### Author Response · Authors · 2021-11-26
> > **Follow up**
> >
> > Dear reviewer 2JvQ, we hope our responses have addressed your valuable comments. Could you please let us know if any further clarification is needed? We would sincerely appreciate the chance to interact before the deadline of the discussion period.

---

> > > ### Comment · Reviewer_2JvQ · 2021-11-29
> > > **Thanks for your response**
> > >
> > > You have addressed my comments. Thank you!

---

> > > > ### Author Response · Authors · 2021-11-29
> > > > **Thanks for acknowledging our response!**
> > > >
> > > > We appreciate the reviewer for acknowledging our response!

---

### Official Review · Reviewer_i87P · 2021-11-02

**Correctness:** 3
**Technical Novelty And Significance:** 2
**Empirical Novelty And Significance:** 2
**Recommendation:** 3
**Confidence:** 4

**Main Review:**

Strength:

1. The paper provides a relatively complete analysis of the Wasserstein and Sinkhorn policy optimization, including closed-form policy updates, performance improvement bound, actor-critic algorithm, and experiments on popular instances.

Weakness:

1. The theoretical advances seem quite limited. The closed-form policy update in Theorem 1 and Theorem 3 should follow from existing duality results on Wasserstein and Sinkhorn optimization (for example, [1] and [2]). It is unfortunate that such connections are not recognized in the paper. Theorem 2 is a simple consequence of the well-known performance difference lemma. Theorem 4 is sort of expected, as many existing results on the relationship between Wasserstein distance and Sinkhorn distance. In addition, it is unclear whether SPO has a performance improvement at each iteration for any fixed entropic regularization parameter.

2. Computational-wise, I am wondering how the policy update in (WPO) and (SPO) can be implemented when the state/action space is large, and the computation of the parameters $\beta_t$ seems also time-consuming whereas I did not find a technically sound argument for the heuristic choices of $\beta$.

3. The experiment results are not convincing enough to demonstrate clear advantages of WPO or SPO.

(1) There is no comparison with existing Wasserstein policy optimization (with parametric classes), such as algorithms in Moskovitz et al. (2020) and Pacchinao et al. (2020). Therefore, it is unclear to me whether it is worth considering a non-parametric policy update, given that is the high computational cost for large spaces.

(2) In most results (for example, Figures 2,3,5), SPO is outperformed by WPO in terms of the reward at the end of the timesteps, although it converges to a suboptimal value faster. It might be helpful to consider a varying entropic regularization parameter.

References.
[1] Blanchet, Jose, and Karthyek Murthy. "Quantifying distributional model risk via optimal transport." Mathematics of Operations Research 44.2 (2019): 565-600.
[2] Wang, Jie, Rui Gao, and Yao Xie. "Sinkhorn Distributionally Robust Optimization." arXiv preprint arXiv:2109.11926 (2021).

**Summary Of The Paper:**

This paper considers Wasserstein and Sinkhorn trust region policy optimization (WPO and SPO) for reinforcement learning. Unlike existing works on WPO, it does not assume a parametric policy class. Theoretically, it shows the performance improvement of WPO at every iteration, and that SPO converges to WPO. It also conducts some experiements to illustrate the advantages of the proposed methods.

**Summary Of The Review:**

The main feature of this paper is considering a non-parametric policy class for Wasserstein policy optimization. However, the theoretical contribution adds marginal to the existing literature, and the numerical findings are not convincing to demonstrate the advantages of the new framework.

---

> ### Author Response · Authors · 2021-11-22
> **Response to Reviewer i87P**
>
> Thanks for your review and comments. We are afraid that the reviewer missed important differences in our contribution from related work, which we hope to clarify below.
>
> **Q1. Connections to Wasserstein/Sinkhorn DRO.**
>
> **Answer:** Thanks for pointing out the two references on Wasserstein/Sinkhorn DRO.  We agree they are (remotely) related, but in no way does such a connection decrease the novelty or significance of our work.
>
> - First, trust region policy optimization is *fundamentally different* from distributionally robust optimization. Constrained policy optimization (e.g., the original TRPO) focuses on finding the *optimistic policy* that falls in a trust region, whereas DRO (e.g., the KL DRO) aims to optimize some worst-case loss given by the *adversarial distribution of unknown parameters* within some ambiguity set. Results in the DRO literature do not directly translate to the policy optimization framework. Many of the state-of-art constrained policy optimization papers (e.g., Abdolmaleki et al., 2018, Peng et al., 2019), which also share a similar closed-form as KL DRO based on duality, did not refer to the DRO literature.
>
> - To the best of our knowledge, our work provides the first and explicit closed-form updates for policy optimization based on Wasserstein and Sinkhorn trust regions. Our results on the monotonic policy improvement and relationship between Sinkhorn and Wasserstein policy optimization, in hindsight, may look ``sort of expected" to the reviewer, but we believe they are novel and meaningful in the context of reinforcement learning.
>
> - Lastly, we kindly point out that the Sinkhorn DRO paper ([2]) brought up by the reviewer should be considered as an independent work: it was recently posted online (09/24/2021) and was not available when we wrote our paper.
>
> **Q2: Computational complexity and choice of $\beta_t$.**
>
> **Answer:**
> *We stress that the proposed method is not intended to address large-scale RL problems nor replace existing heuristic deep RL solvers. But instead, we demonstrate efficient implementations and benefits of using Wasserstein/Sinkhorn divergences and non-parametric policy class, which could be a viable and robust alternative for solving RL problems with moderate size, as evidenced in the numerical experiments.*
>
> As shown in Theorem 2, monotonic improvement of the algorithm holds for any choice of $\beta_t$, so there is no necessity in computing the optimal dual at every iteration. If we choose $\beta_t$ as a time-dependent series, the per-iteration computation cost is $O(N_{A}N_{S})$, where $N_{A}$ and $N_{S}$ are the number of actions and states to perform policy updates. If we set $\beta_t$ as the dual optimizer, this will require some additional cost to run gradient descent to solve the one-dimensional dual formulation. In our experiment, we set $\beta_t$ to be the dual optimizer only in the first few iterations and we use a decaying $\beta_t$ afterward. Therefore, the average computation cost per policy update step is $O(N_{A}N_{S})$.
>
> In implementation, for large state spaces, we update policy for a random subset of states; for large action spaces, we discretize actions. More details can be found in Appendix H.
>
> **Q3(a) Comparison with existing Wasserstein policy optimization with parametric class.**
>
> **Answer:**
>  The reviewer might have missed several key differences in our work from Moskovitz et al. (2020) and Pacchinao et al. (2020).
>
> - These two works focus on Wasserstein distance over the behavioral distributions of policies to regulate behavioral proximity, whereas we directly look at the proximity of the policies.
>
> - These two works require complicated subroutines (which are also expensive) to approximately compute the policy parameter update under the Wasserstein distance, whereas we present simple closed-form policy updates.
>
> - Monotonic policy improvement is not always guaranteed in these two works due to the approximation error induced when computing the parameter update.
>
> Hence, we believe that it is worth considering a non-parametric policy update, from both algorithmic and theoretic viewpoints. Our experiments also clearly demonstrate the benefits of using non-parametric policy compared to TRPO and PPO which limit policies to a certain parametric distribution function.
> As the reviewer suggested, we conduct new experiments and provide direct comparisons with parametric WPO (Pacchinao et al., 2020) in Figure 6.  For a fair comparison, we run experiments on several continuous control tasks (as for tabular domains, using parametric policy clearly has no advantage). We show that our WPO and SPO outperform the parametric WPO on several OpenAI Gym tasks.
>
> **Q3(b) Experiments for varying entropic regularizer.**
>
> **Answer:**
> Thanks for the nice suggestion. We updated Figure 1 with several varying $\lambda$. We observe a similar result: the choice of $\lambda$ can effectively adjust the trade-off between convergence and final performance.

---

> > ### Author Response · Authors · 2021-11-26
> > **Follow up**
> >
> > Dear reviewer i87P, we hope our responses have addressed your valuable comments. Could you please let us know if any further clarification is needed? We would sincerely appreciate the chance to interact before the deadline of the discussion period.

---

> > > ### Comment · Reviewer_i87P · 2021-11-28
> > > **The rebuttal raises more concerns than it answers**
> > >
> > > Thank you for your responses. Unfortunately, I am not able to raise my score because I still have many questions :(
> > >
> > > - Regarding Q1:
> > >
> > > The subproblem in the trust region policy optimization and distributionally robust optimization has a similar form, if not identical, and therefore, mathematically, the results in DRO apply directly to the subproblem reformulation in TRPO. In this respect, the methodological advances in Theorem 1 and Theorem 3 are quite limited. Note that the reformulation involving KL divergence has been studied in statistics long before KL DRO is introduced, so I think it is fine that TRPO based on KL did not cite KL DRO papers but some other statistical literature. On the other hand, to my best knowledge, studies on Wasserstein DRO are very recent but have been recognized by RL literature, e..g, [1]. Therefore, I think the authors should compare the DRO literature in a more careful way — it is not a good excuse to not compare existing literature because some others did not.
> > >
> > > [1] Abdullah, Mohammed Amin, et al. "Wasserstein robust reinforcement learning." arXiv preprint arXiv:1907.13196 (2019).
> > >
> > > As I mentioned previously, the policy improvement of Wasserstein TRPO is a simple corollary of the existing result of KL TRPO using the performance improvement lemma. I do not think this is a significant contribution. In addition, I did not see the response to my previous question on the performance improvement result for Sinkhorn TRPO.
> > >
> > > - Regarding Q2:
> > >
> > > If my understanding of the authors' response is correct, there is an inconsistency between the theoretical results and the empirical experiments: the theoretical results in Section 3 concern with a Wasserstein/Sinkhorn neighborhood, whereas the empirical experiments mostly consider a Wasserstein/Sinkhorn penalty. If so, why not simply consider a Wasserstein/Sinkhorn penalty in the first place and tune the penalty parameter properly? I believe the theoretical analysis would be simpler.
> > >
> > > - Regarding Q3:
> > >
> > > The updated Figure 1 does not show a clear advantage of SPO over WPO in terms of the final performance, which makes me puzzled. If we compare the policy updates (WPO) and (SPO), the latter seems to encourage exploration (due to the entropic regularization) and is less likely to be trapped in a locally optimal policy, but all numerical experiments in Figure 1 (as well as Figure 2) indicate that SPO has a worse final performance compared to WPO. I wonder if the authors can provide some explanation.
> > >
> > > The added comparison with BGPG from Pacchinao et al. (2020) in Figure 6 indicates that BGPG is significantly worse than many benchmarks including TRPO, however, the experiments in Pacchinao et al. (2020) showed that BGPG outperforms TRPO in testing environments like Hopper and Walker2d which is also considered in this paper (see also Moskovitz et al. (2021) for similar conclusions including comparisons with PPO and TRPO). I am wondering why there is such an apparent contradiction and if the conclusion from the numerical experiments is robust.

---

> > > > ### Author Response · Authors · 2021-11-29
> > > > **Thank you for your response_part A**
> > > >
> > > >
> > > > We thank the reviewer for acknowledging our response and the follow-up questions, which we further clarify below.
> > > >
> > > > **Q1:** We would be happy to further expand our discussion on DRO literature and its connection to our algorithms. Note that the reference [1] pointed out by the reviewer is about applying Wasserstein DRO framework to RL, thus of course closely relevant to DRO, but is quite distinct from constrained policy optimization.
> > > >
> > > >
> > > > However, the reviewer seems to have completely ignored our main algorithmic contributions to the reinforcement learning literature -- we introduced two new RL algorithms based on Wasserstein and Sinkhorn policy optimization that are provably efficient in theory and practice. Our work is not just about deriving closed-form updates of extreme distributions under different divergence.  Also, we cannot agree with the reviewer that the existing DRO results can directly apply here simply because both used Lagrangian duality. (Could the reviewer provide specific pointers of existing results in the literature that yields our Theorems 1 and 3?) We sincerely hope that the reviewer could consider re-evaluating our contribution in the context of RL.
> > > >
> > > >
> > > > Regarding the proof of the monotonic performance improvement, with all respect, we cannot agree with the reviewer that simple means insignificant or not important. To the best of our knowledge, there is no literature on constrained policy optimization with Wasserstein metric that has shown their monotonic performance improvement. So we believe this result is significant and meaningful in RL context.
> > > >
> > > > Regarding the convergence of WPO and SPO, the monotonic performance improvement of WPO also implies the monotonic performance improvement of SPO with a large $\lambda$.
> > > >
> > > >
> > > > **Q2:** We are not sure what ``Wasserstein/Sinkhorn penalty" the reviewer refers to. We adopt a Wasserstein/Sinkhorn constraint/neighborhood in both theoretical results and empirical experiments. If the reviewer refers to the Lagrangian multiplier $\beta$, we have provided the guidelines to determine $\beta$ value in both an optimal and a time-varying way. There is no need to tune $\beta$ blindly. If we simply penalize the trust region constraint, it is hard to determine or tune the penalty parameter to its optimal value. Nevertheless, our framework can apply to the case of penalizing the trust region, with $\beta$ being the penalty parameter. However, we believe our work is more general from both theoretical and experimental viewpoints and is able to address both constrained and penalized cases.

---

> > > > > ### Author Response · Authors · 2021-11-29
> > > > > **Thank you for your response_part B**
> > > > >
> > > > >
> > > > > **Q3 (a). WPO vs SPO** As we explicitly stated in the paper (see our abstract and Section 6.1), SPO has a faster convergence than WPO but has no advantage over WPO in terms of the final performance. Based on our theory, SPO can approximate WPO and match its performance when $\lambda\to\infty$. This exactly aligns with our observation in the numerical experiments.
> > > > >
> > > > >
> > > > > The reviewer might have some misinterpretation of the entropy regularization introduced in the Sinkhorn distance.  The entropy regularization is imposed to penalize the coupling of two policies, in order to smooth out the Wasserstein distance and mitigate the computational burden; see our discussion in ``Sinkhorn Divergence" subsection on Page 3. This is fundamentally different from the purpose of the commonly used entropy regularization technique (imposed on the stochastic policy itself) in policy optimization to encourage exploration.
> > > > >
> > > > >
> > > > > **Q3 (b). BGPG vs TRPO/PPO** One of the reasons might come from the discrepancy of the TRPO/PPO benchmark used in our paper and in Pacchinao et al., (2020).
> > > > >
> > > > > - We use TRPO/PPO implementation from OpenAI baseline with customized neural network sizes ([400, 300]). Our results of TRPO/PPO on Walker2d and Hopper are consistent with the original TRPO paper (Schulman et al., 2015, Figure 4) and the original PPO paper (Schulman et al., 2017, Figure 3), which are approximate 2200 for Hopper and 3000 for Walker2d.
> > > > > - In Pacchinao et al., (2020), their reported TRPO/PPO has much worse performance than the one shown in our paper and original TRPO/PPO papers: TRPO/PPO only reaches 0 for Hopper and 500 for Walker2d.
> > > > >
> > > > > - For BGPG,  we adopt the same implementation as Pacchinao et al., (2020) based on the released code *https://github.com/behaviorguidedRL/BGRL*. We only tune the neural network sizes to match our setup;  the other settings are default in their codebase. Our results of BGPG method are similar to the ones shown in Pacchinao et al., (2020) in Walker2d.
> > > > >
> > > > > - For the reviewer's reference, the implementation of our WPO/SPO can be found in https://github.com/efficientwpo/EfficientWPO.
> > > > >
> > > > > We suspect the poor performance of BGPG might relate to the sensitivity of BGPG in terms of the initialization and stepsizes. As we argued in the paper, since parametric policies in constrained policy optimization are not convex in the distribution space, optimizing over such policies may easily result in local movements in the action space and thus be stuck in a suboptimal policy. On the contrary, our proposed method does not have this issue. Since the policy learned by WPO/SPO is not confined to the scope of parametric functions, the new policy is always the one
> > > > > that can maximize the advantage function within the trust region.
> > > > >  We believe this experiment further justifies the importance of considering non-parametric policy update in our algorithms.

---

> > > > > > ### Comment · Reviewer_i87P · 2021-11-29
> > > > > > **Further questions and clarifications**
> > > > > >
> > > > > > - WPO vs SPO
> > > > > >
> > > > > > To clarify, I did not look directly at the entropic regularization in the SPO formulation, but compared the policy update rules in the formulas (WPO) and (SPO) in the submission. (WPO) tends to put all probability in the argmax, but (SPO) put the probability over all actions, which is what I meant by "it seems to encourage exploration and escape from local optimum". Since the RL objective is non-convex in the policy, WPO may converge to a locally optimal policy (at least there's no result to show its global optimality to my best knowledge). Given the exploration capability of SPO, there's a hope that SPO might improve the convergence of WPO, so I am curious why this is not the case according to the authors' experiments.
> > > > > >
> > > > > > - BGPG vs TRPO/PPO
> > > > > >
> > > > > > The mentioned references use different experimental setups. Since the numerical observations are different under different experimental settings, I am worried if it is safe to claim the empirical advantage of WPO based on the current numerical results. Definitely, more comparisons and robustness checks are needed. It would be helpful to point out regimes where WPO/SPO outperforms other benchmarks.
> > > > > >
> > > > > > - On policy parameterization.
> > > > > >
> > > > > > I am confused about the authors' claim that a non-parametrized policy is good, as how the policy is parameterized has a significant impact on the convergence rate. For one thing, as I mentioned in the other follow-up, on a finite action space, it is known that parameterization can affect the convergence behavior, for example, the non-parameterized policy (also called the direct policy parameterization in [3]) is often worse than soft-max parameterization, where the latter often has global optimality guarantees with a faster convergence rate. On the other hand, there is no global guarantee of the WPO in this paper, although the trust-region subproblem can be solved to global optimality.

---

> > > > > > > ### Author Response · Authors · 2021-11-29
> > > > > > > **Follow-up Responses**
> > > > > > >
> > > > > > > > Given the exploration capability of SPO, there's a hope that SPO might improve the convergence of WPO, so I am curious why this is not the case according to the authors' experiments.
> > > > > > >
> > > > > > > The reviewer might be right, but we have not observed potential benefits of better exploration of SPO in our experiments.
> > > > > > >
> > > > > > > >  Definitely, more comparisons and robustness checks are needed. It would be helpful to point out regimes where WPO/SPO outperforms other benchmarks.
> > > > > > >
> > > > > > >  We completely agree and will add these discussions in our revision.
> > > > > > >
> > > > > > > > I am confused about the authors' claim that a non-parametrized policy is good, as how the policy is parameterized has a significant impact on the convergence rate. For one thing, as I mentioned in the other follow-up, on a finite action space, it is known that parameterization can affect the convergence behavior, for example, the non-parameterized policy (also called the direct policy parameterization in [3]) is often worse than soft-max parameterization, where the latter often has global optimality guarantees with a faster convergence rate. On the other hand, there is no global guarantee of the WPO in this paper, although the trust-region subproblem can be solved to global optimality.
> > > > > > >
> > > > > > > The reviewer is absolutely right that different parametrization will lead to different convergence guarantees. We did not claim that tabular parametrization is always better. For discrete domains and small continuous control tasks, it could be more advantageous than considering a parametric policy class based on neural networks; the latter has no performance guarantee in general.   This is also evidenced in our experiments.
> > > > > > >
> > > > > > > The study of global optimality guarantees for policy gradient methods is still an active research field. It is definitely interesting to explore softmax parametrization in WPO setting and its global convergence rate. We will leave for future work and thank the reviewer for the suggestion.

---

> > > > > ### Comment · Reviewer_i87P · 2021-11-29
> > > > > **Follow-up questions and clarifications**
> > > > >
> > > > > - Related reference in DRO:
> > > > >
> > > > > For the Wasserstein policy update, Theorem 1 can be viewed as a special case of Theorem 1(b) in [1] which provides the distribution update on a metric space with general transport cost functions, which covers the discrete space and cost matrix M considered by this submission.
> > > > >
> > > > > For the Sinkhorn policy update, Theorem 3 can be viewed as a special case of Remark 5 in [2] which provides the distribution update also for metric spaces and general cost functions.
> > > > >
> > > > > [1] Blanchet, Jose, and Karthyek Murthy. "Quantifying distributional model risk via optimal transport." Mathematics of Operations Research 44.2 (2019): 565-600.
> > > > > [2] Wang, Jie, Rui Gao, and Yao Xie. "Sinkhorn Distributionally Robust Optimization." arXiv preprint arXiv:2109.11926 (2021).
> > > > >
> > > > > As the authors said, the contribution should be on the algorithmic part, but theoretically, it is essentially an application of existing results of DRO, and the authors should be more cautious in claiming their contribution on deriving closed-form policy updates based on the Lagrangian duality. Given that Theorem 1 and Theorem 3 are major results in the current presentation, I still think their contribution is limited.
> > > > >
> > > > > - A technical issue in Theorem 1 and (WPO).
> > > > >
> > > > > When the argmax in (WPO) is not unique, why an arbitrary tie-breaking rule leads to a feasible policy in the trust region? From the proof of Theorem 1, it is unclear to me why the updated policy satisfies (10a).
> > > > >
> > > > > - On the performance improvement.
> > > > >
> > > > > Theorem 2 is really a straightforward application of the performance difference lemma (15a) and the policy update (WPO). The right-hand side of (7) depends on a quantity $\hat{k}_s^{\pi_t}(\beta_t,j)j$, which is hard to lower bound other than non-negativity.
> > > > >
> > > > > The authors claim in the paper that the performance improvement bound is tighter than the KL-based trust region, but natural policy gradient with softmax policy parameterization has been shown to converge globally [3,4]. Btw, this also highlights that parameterization has a direct impact on the convergence--it is not the case that the direct parameterization is always good as claimed in the reviewers' rebuttal.
> > > > >
> > > > > In my initial feedback, I asked about the performance improvement of the Sinkhorn update for any FIXED entropic regularization parameter, but I still do not see a response on that. The convergence with large entropic regularization parameter is also vague, because Theorem 4 in the paper only shows Sinkhorn converges to Wasserstein in the limit, but there is no guaranteed bound for any finite lambda.
> > > > >
> > > > > [3] Agarwal, Alekh, et al. "On the theory of policy gradient methods: Optimality, approximation, and distribution shift." Journal of Machine Learning Research 22.98 (2021): 1-76.
> > > > > [4] Bhandari, Jalaj, and Daniel Russo. "A note on the linear convergence of policy gradient methods." arXiv preprint arXiv:2007.11120 (2020).
> > > > >
> > > > > - On the Wasserstein/Sinkhorn penalty problem and the numerical experiments
> > > > >
> > > > > If $\beta$ is chosen in advance instead of solving to optimality, then the problem can be viewed as the reformulation for a Wasserstein/Sinkhorn penalty problem, namely, the problem where the trust-region constraint appears as a soft penalty in the objective.
> > > > >
> > > > > For the numerical experiments in Sections 6.2, 6.3, 6.4, did you actually solve the optimal $\beta$? My understanding is that for most iterations, it just solves the Wasserstein/Sinkhorn penalty problem except for the first few iterations. Since solving optimal $\beta$ requires additional time, why not just consider the Wasserstein/Sinkhorn penalty problem theoretically in the first place?

---

> > > > > > ### Author Response · Authors · 2021-11-29
> > > > > > **Follow-up Responses**
> > > > > >
> > > > > > We are grateful for the reviewer's spirit of critical thinking, which in fact,  reinforces our confidence in the contribution of our work. Before addressing your remaining concerns, we would like politely ask the reviewer to please read through our previous responses so that we don't have to repeat the same questions over and over.
> > > > > >
> > > > > >
> > > > > > > Theorem 1 is a special case of Theorem 1(b) in Jose and Murthy (2019)?
> > > > > >
> > > > > > Theorem 1(b) in Jose and Murthy (2019) merely provides a KKT condition for the optimal primal-dual solutions in general setting, which does not yield any explicit close-form solutions of the optimal solutions. Does the reviewer agree the difference?
> > > > > >
> > > > > >
> > > > > > > Theorem 3 can be viewed as a special case of Remark 5 in Jie, Gao and Xie (2021)?
> > > > > >
> > > > > > As we already mentioned in the last response, it is unfair to compare our result to a concurrent work that was released on arXiv after our submission.
> > > > > >
> > > > > >
> > > > > > > Given that Theorem 1 and Theorem 3 are major results in the current presentation, I still think their contribution is limited.
> > > > > >
> > > > > >
> > > > > > This is very inaccurate. Our major results are not limited to Theorem 1 and Theorem 3. We remind the reviewer again that in this paper, we introduced two natural extensions of TRPO, which are practically efficient and outperform many benchmarks on various tasks. To the best of our knowledge, these two variants have not been yet studied in the RL literature.
> > > > > >
> > > > > >
> > > > > > > The authors claim in the paper that the performance improvement bound is tighter than the KL-based trust region, but natural policy gradient with softmax policy parameterization has been shown to converge globally.
> > > > > >
> > > > > > We don't understand the point. How does the global convergence of NPG contradict with the performance improvement bound of WPO?
> > > > > >
> > > > > >
> > > > > > > Btw, this also highlights that parameterization has a direct impact on the convergence--it is not the case that the direct parameterization is always good as claimed in the reviewers' rebuttal.
> > > > > >
> > > > > > This is again a false accusation. We did not claim that direct parametrization is always better in the rebuttal. We stated that policy optimization under parametric policy class is in general heavily nonconvex and might stuck in local suboptimal solutions. Our numerical experiments show inferior performance of BGPG using parametric policies compared to our method without parametrization on small continous control tasks.
> > > > > >
> > > > > > > I asked about the performance improvement of the Sinkhorn update for any FIXED entropic regularization parameter, but I still do not see a response on that.
> > > > > >
> > > > > >
> > > > > > As we mentioned before, there is no guarantee on monotonic performance improvement of the Sinkhorn updated with abitrarily fixed entropic regularization (Think about the case when $\lambda\to 0$).  That's why we require $lambda$ to go to infinity or sufficiently large.
> > > > > >
> > > > > >
> > > > > > > The convergence with large entropic regularization parameter is also vague, because Theorem 4 in the paper only shows Sinkhorn converges to Wasserstein in the limit, but there is no guaranteed bound for any finite lambda.
> > > > > >
> > > > > > The rate of convergence of $F_{\lambda}(\beta)$ to $F(\beta)$ is indeed of order $1/\lambda$, which can be found in Equation (39) of Appendix F. See our response to Reviewer A8WK.
> > > > > >
> > > > > >
> > > > > > > Why not just consider the Wasserstein/Sinkhorn penalty problem theoretically in the first place?
> > > > > >
> > > > > >
> > > > > > Our motivation from the very beginning is to extend TRPO with Wasserstein/Sinkhorn divergences. Secondly, while directly considering penalty problems makes the theory easier, it is less general. As the reviewer already explained, our results essentially encompasses both cases, depending on how we select $\beta$.  Lastly, in practice, as we already shown in the experiments (see Figure 1), using the optimal $\beta$ (even just in the begnning) performs better than fixing $\beta$.

---

> ### Comment · Reviewer_i87P · 2021-11-30
> **Summary and clarification of major concerns**
>
> To avoid further confusion, let me highlight my major concerns about the claimed contribution.
>
> 0. *(From abstract) Instead of restricting the policy to a parametric distribution class, we directly optimize the policy distribution...(From rebuttal)...demonstrate efficient implementations and benefits of using Wasserstein/Sinkhorn divergences and non-parametric policy class.*
>
> The authors focus on finite action-state spaces in theory and tabular/moderate-size in experiments. In these settings, Agarwal et al. (2019) has proved global convergence for algorithms like nonparametric class and KL-based method. As such, justification of the proposed approach is necessary. Unfortunately, as my point 2 below, theoretically, I cannot see a clear advantage of Wasserstein with non-parametric class other than being a natural extension of TRPO; and as my point 3 below, the numerical experiments are not yet convicing. Surely, we may not expect both aspects in a conference paper, but the current results need improvement as detailed below.
>
> 1. *Algorithms: We develop close-form expressions of the policy updates for both WPO and SPO based on the corresponding optimal Lagrangian multipliers of the trust-region constraints. In particular, the optimal Lagrangian multiplier of SPO admits a simple form and can be computed efficiently. A practical on-policy actor-critic algorithm is proposed based on the derived expressions of policy updates and advantage value function estimation.*
>
> - No doubt that the algorithm in Section 5 is new.
>
> - The closed-form expressions in WPO is not new. For WPO, $\pi^*$ in Theorem 1(b) and Remark 2 in Blanchet and Murthy (2019) gives a similar expression. Note that the support of $\pi^*$ is the argmax set.
>
> - I doubt the technical correctness of Theorem 1 since it is unclear to me why the policy update (WPO) satisfies the feasibility constraint (10a). Remark that Theorem 1(b) in Blanchet and Murthy (2019) did not claim an arbitrary tie-breaking rule.
>
> 2. *Theory: We theoretically show that WPO guarantees a monotonic performance improvement through the iterations, even with non-optimal Lagrangian multipliers, yielding better and more robust guarantee compared to that using KL divergence. Moreover, we prove that SPO converges to WPO as the entropic regularizer diminishes.*
>
> - The performance improvement of WPO is new, although it is a straightforward application of the performance difference lemma and the formula of WPO policy update. It may be improved by deriving a bound on the second term in (7).
>
> - I am not sure whether WPO is “yielding better and more robust guarantee compared to that using KL divergence”. In the paper, the authors argue that the second term in (7) is likely to be positive and thus is better than KL by referring to some literature that did not provide a positive lower bound on the improvement in KL. However, there are results showing a strictly positive lower bound for natural policy gradient, for example, Lemma 5.2 in Agarwal et al. (2019) (which in fact proves global convergence).
>
> - The convergence of SPO to WPO is new, but no performance improvement of SPO for finite and large $\lambda$ is derived. The authors responded that they are able to prove it using proof of Theorem 3. But what's unclear to me is how the convergence of the SPO policy to WPO (even with an explicit rate) implies the performance improvement of SPO for large $\lambda$, precisely?
>
> 3. *Experiments: A comprehensive evaluation with several types of testing environments including tabular domains and robotic locomotion tasks demonstrates the efficiency and effectiveness of WPO and SPO. Compared to state-of-art policy gradients approaches using KL divergences such as TRPO and PPO, our methods achieve better sample efficiency, faster convergence, and improved final performance. Our numerical study indicates that by properly choosing the weight of the entropic regularizer, SPO achieves a better trade-off between convergence and final performance than WPO.*
>
> - Based on the results in the initial version and the updated results in the rebuttal, it is not clear enough to see the benefit of WPO/SPO with a non-parametric policy class: when the proposed framework works best, whether the performance gain is due to a Wasserstein/Sinkhorn trust-region or a non-parametric class. This may be improved by comparing KL TRPO with non-parametric policies and WPO with parametric policies (note that WNG in Moskovitz et al. 2021 Eq.(8) applies to RL with or without a behavior map though they focused on the former).
>
> - I am concerned with the robustness of the numerical conclusions, because of the inconsistency with existing works. This can be improved by expanding the experiments.

---

> > ### Author Response · Authors · 2021-11-30
> > **Follow up responses to your comments**
> >
> > # Algorithms
> >
> > >>> The closed-form expressions in WPO/SPO are not new.
> >
> > **A:** We have already answered this question in our last responses: (a) Theorem 1(b) of Blanchet and Murthy (2019) only provides a general KKT condition, with *NO* closed-form solution given; while our Theorem 1 provides a *tractable closed-form solution*. (b) We have already cited Wang et al. (2021) and discussed the similarity and differences between our work and DRO literature including Wang et al. (2021) in our new version. And again, we do not think this concurrent work should weaken our contribution.
> >
> > >>> Remark that Theorem 1(b) in Blanchet and Murthy (2019) did not claim an arbitrary tie-breaking rule.
> >
> > **A:** Remark 3 of Theorem 1(b) on page 8 of Blanchet and Murthy (2019) assumes a unique optimizer. Therefore there is no need for them to consider the tie-breaking rule. Their theorem can be viewed as a *special case* of a *part* of our Theorem 1 proof. Our Theorem 1 is more general.
> >
> > # Theory
> >
> > >>> However, there are results showing a strictly positive lower bound for natural policy gradient, for example, Lemma 5.2 in Agarwal et al. (2019) (which in fact proves global convergence).
> >
> > **A:** The reviewer's claim is wrong. Lemma 5.2 in Agarwal et al. (2019) reads:
> > $V^{t+1}(\mu) - V^{t}(\mu) \ge \frac{(1-\gamma)}{\eta}E_{s \sim \mu} \log Z_t (s) \ge 0$,
> > which is *NOT* a strictly positive lower bound. We also remind the reviewer that NPG$\neq$TRPO!
> >
> > >>> It may be improved by deriving a bound on the second term in (7)
> >
> > **A:** Our second term in Equation (7) is strictly positive in general, thus our performance improvement bound is better than that of NPG shown in Lemma 5.2 in Agarwal et al. (2019). Here we provide the derivation:
> > When $\pi_{t+1} \ne \pi_{t}$ (which generally holds true since policy needs to be updated), there exists a $j'$ such that $\pi_t(a_{j'}|s) > 0$ and ${k}_s^{\pi_t}(\beta_t,j') \ne {j'}$, which means ${\sum}_{j=1}^N \pi_{t}(a_j|s) M_{{k}^{\pi_t}_s(\beta_t, j)j} \geq \pi_t(a_{j'}|s)M_{k_s^{\pi_t}(\beta_t,j')j'} > 0$ and thus the second term in Equation (7) is strictly positive in general.
> >
> > >>> no performance improvement of SPO for finite and large $\lambda$ is derived. The authors responded that they are able to prove it using proof of Theorem 3.
> >
> > **A:** We *NEVER* claimed that we are able to prove performance improvement of SPO. Instead, in our last response, we stated that "there is no guarantee on monotonic performance improvement of the Sinkhorn updated with arbitrarily fixed entropic regularization". We would like the reviewer to carefully read our response.
> >
> > # Experiments
> >
> > >>> This may be improved by comparing KL TRPO with non-parametric policies and WPO with parametric policies (note that WNG in Moskovitz et al. 2021 Eq.(8) applies to RL with or without a behavior map though they focused on the former).
> >
> > **A:** A comparison of KL TRPO with non-parametric KL policy update has already been studied in AWR (Peng et al., 2019) and MPO (Abdolmaleki et al., 2018). We have added comparisons of our WPO to parametric WPO (Pacchiano et al., 2020) in Figure 6 of our updated paper. We are willing to further compare with MNG (Moskovitz et al. 2021) in our final version.
> >
> > >>> I am concerned with the robustness of the numerical conclusions, because of the inconsistency with existing works. This can be improved by expanding the experiments.
> >
> > **A:** We have already answered this question in our previous response.

---

> > > ### Comment · Reviewer_i87P · 2021-11-30
> > > **Further questions**
> > >
> > > The responses are not very helpful in general, please calm and think in more depth. Let me provide more explanations below.
> > >
> > > First, I would encourage the authors to read Blanchet and Murthy (2019) more carefully and recognize their contributions. Their Theorem 1(b) describes the support of the transport plan between the policy in the current iteration and the next iteration, and their Remark 2 studies the structure and explicitly mentions that *the support of the new policy is on the argmax set*. Moreover, they do not assume uniqueness in their general Theorem 1(b) but only Remark 3 as a discussion. Their result holds for general metric spaces and transport cost structure, so it is dangerous to claim their theorem is a special case of yours. Last but not the least, please explain why *your update rule in (WPO) satisfies the feasibility constraint*.
> > >
> > > Second, regarding Agarwal et al. (2019) Lemma 5.2, your argument for Wasserstein applies to NPG as well: the lower bound $\\mathbb{E}_{\\mu} \\log Z_t(s)$ is *strictly positive when the two iterates are different* using the equality condition in Jensen's inequality. Let alone that they have shown global convergence while it is unclear whether WPO has global convergence. Moreover, as I quoted, "it is known in the literature that in the limit of small stepsizes, NPG and TRPO updates are closely related (e.g. see Schulman et al. [2015], Neu et al. [2017], Rajeswaran et al. [2017])".
> > >
> > > Third, regarding the performance improvement of SPO, my question was on the guarantee of the Sinkhorn updates with *fixed and large* $\lambda$. Please clarify whether you can prove it or not using the convergence of SPO to WPO.

---

> > > > ### Author Response · Authors · 2021-11-30
> > > > **We believe we have already addressed all rational concerns of the reviewer. Thanks again for your review.**

---

> > > > > ### Comment · Reviewer_i87P · 2021-11-30
> > > > > **Score unchanged due to unexplained technical concerns and lack of comparisons with literature**
> > > > >
> > > > > After several rounds of discussions, I will not be able to raise my score, unfortunately. This is mainly because of the technical concerns about Theorem 1 which the authors did not respond to, and a lack of comparison with literature, especially Blanchet and Murthy (2019) which the authors did not recognize.

---

> > ### Comment · Program_Chairs · 2021-12-02
> > **Comparison with concurrent work**
> >
> > Quoting the reviewer guidelines (https://iclr.cc/Conferences/2022/ReviewerGuide):
> > "... if a paper was published (i.e., at a peer-reviewed venue) on or after June 5, 2021, authors are not required to compare their own work to that paper."
> >
> > Therefore, the request to compare to Wang et al. (2021) should be withdrawn, because that paper appeard on arXiv only a few days before the ICLR deadline.

---

> > > ### Comment · Reviewer_i87P · 2021-12-02
> > > **Thanks for pointing this out**
> > >
> > > I have modified my point 1(2). My other comments remain unchanged.

---

### Official Review · Reviewer_PvnA · 2021-11-05

**Correctness:** 3
**Technical Novelty And Significance:** 2
**Empirical Novelty And Significance:** 3
**Recommendation:** 6
**Confidence:** 3

**Main Review:**

The paper is well-organised and present a new way of incorporating the Wasserstein distance within policy optimisation algorithms.
The theoretical results look correct at first glance although I admit didn't check them carefully in the Appendix.

It would be nice for theorem 1 to be self-contained: some of the variables are defined earlier in the text like M or beta but it would ease the reading to define them in the theorem. The presentation of the theoretical results are a bit hard to follow so adding a few explanatory sentences about the importance of each terms would also be helpful.

Could the authors also comment on the computational complexity of the method?

In terms of experiments, the domains considered seem to show the benefits of the method. I believe stronger tasks would make the paper stronger, in particular continuous control ones or at least showing results for a few more domains.

Related Work: "Wasserstein-like metrics have only been recently studied in the context of reinforcement learning." Please note that the Wasserstein metric has been used in RL since at least 2012 with the introduction of bisimulation metrics https://arxiv.org/pdf/1207.4114.pdf, and recent work havent only used it for imitation learning but also for generalization https://arxiv.org/pdf/2101.05265.pdf https://arxiv.org/abs/2102.01514 https://arxiv.org/abs/2006.10742.
I believe the Wasserstein and Sinkhorn metrics have also been used in the distributional RL literature so it might be nice to discuss the similarity of ideas in both areas.

Typos:
Wassersteim
*the* Wasserstein metric, *etc.* : please dont use "etc" and add all necessary details.

**Summary Of The Paper:**

This paper proposes to use two extensions of the TRPO algorithm relying on the Wasserstein distance and the Sinkhorn divergence which dont require to explicitly specify a distribution for the policy. The authors provide a theoretical analysis giving a closed form policy update for their two methods and a performance improvement bound in the case of Wasserstein policy optimisation. They evaluate their methods empirically on tabular domains (Taxi, Chain and Cliff Walking) and on some discrete locomotion tasks (Cartpole, Acrobot). They find that their method outperform TRPO and PPO while being more sample-efficient, and converging faster.

**Summary Of The Review:**

This paper provides a new way of optimising the policy distribution relying on the Wasserstein and Sinkhorn distances. The methods is theoretically grounded so I would recommend an accept but I believe the authors would need to evaluate their method on more domains to make the paper stronger.

---

> ### Author Response · Authors · 2021-11-22
> **Response to Reviewer PvnA**
>
> Thank you for your positive comments and valuable feedback. We have incorporated all your suggestions in our updated submission. Please see below for our response to your comments and refer to further details in the revision.
>
> **Q1. Presentation of theorems.**
>
> **Answer:**
> Thanks for the suggestion.  In our new version, we restated definitions and added explanations to theorems so that they are more self-contained and easier to follow.
>
> **Q2. The computational complexity of the method.**
>
> **Answer:**
> This is a very good question.  Our overall algorithm applies a general actor-critic framework: the actor follows the proposed WPO or SPO update while the critic follows some TD methods.  The computational complexity depends on (i) the per-iteration computation cost of the policy and critic update and (ii) the iteration complexity of the actor-critic method. Here we mainly discuss the per-iteration computation cost of the policy update, as studies on the iteration complexity of actor-critic framework for constrained policy optimization are limited.
>
> The computation cost of WPO and SPO updates at each iteration depends on the selection of $\beta_t$:
>
> * If $\beta_t$ is chosen arbitrarily and time dependently, the computation cost is $O(N_{A}N_{S})$,   where $N_{A}$ and $N_{S}$ are the number of actions and states to perform policy update.
>
> * If we set $\beta_t$ as the dual optimizer, this will require some additional cost to run gradient descent to solve the one-dimensional dual formulation. But the number of gradient descent steps is considerably small.
>
> In our experiment, we set $\beta_t$ to be the dual optimizer only in the first few iterations and we use a decaying $\beta_t$ afterward.   Therefore, the average computational complexity of a policy update step is $O(N_{A}N_{S})$.  Note that as shown in Theorem 2, we can achieve monotonic performance improvement for any arbitrary $\beta_t\geq 0$, so there is no necessity in computing the optimal dual at every iteration.
>
> **Q3. Experiments with stronger tasks.**
>
> **Answer:**
> Thanks for the suggestion.  We think our experiments already cover a reasonably diverse set of tasks on tabular domains (including Taxi, Chain and Cliff Walking) and discrete locomotion (including Cartpole, Acrobot), that supports the empirical effectiveness of the approach.
>
> Nonetheless, per the reviewer's suggestion, we further extended our experiments to several OpenAI Gym environments with continuous action space, including LunarLander, Hopper and Walker. New results can be found in Section 6.4 on Page 9 in our updated submission.  In particular, we compared our method (after discretizing the action space) with TRPO, PPO, and parametric Wasserstein policy optimization.  Similar results are observed as the discrete action tasks: our WPO and SPO outperform the benchmark algorithms in terms of final performance.
>
> **Q4. Related work.**
>
> **Answer:**
> We thank the reviewer for pointing out the important references on bisimulation metrics and imitation learning. We have added discussions on their connections to our work in the related work section.

---

> > ### Author Response · Authors · 2021-11-26
> > **Follow up**
> >
> > Dear reviewer PvNA, we hope our responses have addressed your valuable comments. Could you please let us know if any further clarification is needed? We would sincerely appreciate the chance to interact before the deadline of the discussion period.

---

### Official Review · Reviewer_A8WK · 2021-11-08

**Correctness:** 4
**Technical Novelty And Significance:** 3
**Empirical Novelty And Significance:** 3
**Recommendation:** 6
**Confidence:** 3

**Main Review:**

The paper is overall well written, the results are well organized and reasonably clear to follow. In terms of novelty, I believe the closed form update parts (Theorem 1 & 3) are reasonably standard primal-dual arguments. However, Theorem 2 is an interesting and significant contribution since it provides theoretical guarantees even when the update $\beta_t$ is not the optimal solution in (6), which circumvent the difficulty in obtaining closed form solution of (6). Theorem 4 also seems important as it gives theoretical guarantee for us to approximate WPO by SPO with large $\lambda$. I would like to discuss a few more questions with the authors:

The upper bound of $\beta^*_{\lambda}$ for SPO in Theorem 3 is of order $1/\delta$, which seems not very good because the bound would be trivial if there is no perturbation to the policy (i.e., $\delta = 0$). Also, the bound for $\beta^*$ for WPO in Theorem 1 seems to be independent of $\lambda$, I would like the authors to discuss some insights behind the different dependency on $\lambda$. Do we expect the bound for $\beta^*_{\lambda}$ is actually also independent of $\delta$. Moreover, it would be also interesting to understand the optimal dependency of $\beta^*_{\lambda}$ as a function of $\lambda$.
In Theorem 4, can we characterize the rate of convergence of $|F_{\lambda}(\beta) - F(\beta)|$? Seems like an upper bound could be $1/\lambda$, but is this correct (or optimal if it is correct)? Also, it would be good if we could understand the convergence of the optimizers in part 2 of Theorem 4.

Small typo in Abstract: ‘extensions of policy optimziation’ -> ‘extensions of policy optimization’.


**Summary Of The Paper:**

This paper studies policy optimization in reinforcement learning with Wasserstein and Sinkhorn trust regions. Compare to the standard TRPO which based on KL-divergence, the proposed WPO and SPO go beyond the parametric policy distribution class. The authors also derive closed form policy update, as well as theoretical performance guarantees for both problems.

**Summary Of The Review:**

This paper presents two policy update frameworks (WPO and SPO) which relax the restriction to parametric policy distribution in the standard setting. Theoretical guarantees are provided. The numerical results also suggest that proposed policy optimization methods outperform the standard TRPO and PPO with better performance and faster convergence.

---

> ### Author Response · Authors · 2021-11-22
> **Response to Reviewer A8WK**
>
> Thank you for your valuable comments and positive reviews. Please see below our answers to your questions.
>
> **Q1. Upper bounds on $\beta_\lambda^\*$ and $\beta^\*$ and their dependencies on $\lambda, \delta$.**
>
> **Answer:**
>
> 1. First, we would like to clarify that the upper bounds of $\beta^\*$ and $\beta_\lambda^\*$ derived in Theorems 1 and 3 are only used to establish the uniform convergence in  Theorem 4.  Having the upper bounds independent of $\lambda$ is actually desired for proving the uniform convergence.
> Note these bounds are not used to approximate the optimizer $\beta^*$ or $\beta_\lambda^\*$, thus are not intended to be tight. We think the current finite bounds $\beta^\* \leq \bar{\beta}$ and $\beta_\lambda^\* \leq 2A^{\max}/\delta$ suffice our needs.
>
> 2. Second, the dependence of the upper bound of $\beta_\lambda^\*$ on $\delta$ is as expected:  recall that $\delta$ denotes the trust region size; when $\delta$ becomes smaller, trust region constraint becomes harder to be satisfied and a larger dual variable $\beta$ is needed (to enforce more penalty). Note that as $\delta\to 0$, i.e., policy should remain the same, in this case we should have $\beta_\lambda^\*=\infty$, which exactly ensures $\pi_\lambda^\*=\pi$ from the update rule in Equation (8).
>
> **Q2. Characterize the convergence rate of $F_{\lambda}(\beta)-F_{\lambda}(\beta)$ and the optimizers.**
>
> **Answer:** The reviewer is correct. The rate of convergence of $F_{\lambda}(\beta)$ to $F(\beta)$ is indeed of order $1/\lambda$, which can be found in Equation (39) of Appendix F. The convergence of optimizers directly follows from properties of epi-convergence. More details on the convergence of optimizers for epi-converging functions can be found in Chapter 7.E of [1].
>
> [1] R. Tyrrell Rockafellar and Roger J. B. Wets. Variational Analysis. Springer, 1998.

---

> > ### Author Response · Authors · 2021-11-26
> > **Follow up**
> >
> > Dear reviewer A8WK, we hope our responses have addressed your valuable comments. Could you please let us know if any further clarification is needed? We would sincerely appreciate the chance to interact before the deadline of the discussion period.

---

> > > ### Comment · Reviewer_A8WK · 2021-11-29
> > > **Post rebuttal decision**
> > >
> > > Thank you for the response to my comments, I decided to keep my original score unchanged.

---

### Author Response · Authors · 2021-11-30
**Summary of our revision**

Dear AC and Reviewers,

Thank you very much again for your valuable suggestions. We would like to highlight a few changes in our new version of the paper:

- In *Related work section on page 2*, we added more discussion on prior RL work using Wasserstein and Sinkhorn distances. We also added connections to DRO (Reviewers PvnA, i87P, and 2JvQ).

- In *Section 6.4 on page 9*, we added experiments on tasks with continuous action space (Reviewer PvnA).

- In *Figure 1 on page 7*, we added experiments on time-varying $\lambda$ for ablation study (Reviewer i87P).

- We added several definitions and explanations to make theorems easier to follow (Reviewer 2JvQ).

Thank you very much.

The Authors

---

### Decision · Program_Chairs · 2022-01-20

**Decision:**

Reject

**Comment:**

This paper proposes two extensions of the TRPO algorithm in which the trust region is defined using the Wasserstein distance and the Sinkhorn divergence. The proposed methods do not restrict the policy to belong to a parametric distribution class and the authors provide
closed-form policy updates and a performance improvement bound for the Wasserstein policy optimization.
The authors provide an empirical evaluation of their approaches on tabular domains and some discrete locomotion tasks, comparing the performance with some state-of-the-art policy optimization approaches.

After reading the authors' feedback and interacting with the authors, the reviewers did not reach a consensus: one of the reviewers votes for rejection, while the other three reviewers are slightly positive.
In particular, the reviewer that voted for rejection raised a number of concerns that have been discussed at length with the authors, who were able to clarify some of the issues, but some of the answers did not satisfy the reviewer.
I went through the paper and I found the paper solid from a technical point of view, but I share some of the reviewers' concerns and I think that the authors should better position their contribution with respect to the state of the art.
Overall, this paper is borderline and I feel it needs still some work to deserve clear acceptance (which I think will be soon).